

# The thermoelectric conversion efficiency problem: Insights from the electron gas thermodynamics close to a phase transition

I. Khomchenko[1,2,3], A. Ryzhov[1], F. Maculewicz[4], F. Kurth[5], R. Hühne[5],
A. Golombek[6], M. Schleberger[6], C. Goupil[7], Ph. Lecoeur[8], A. Böhmer[9],
G. Benenti[2,3,10], G. Schierning[11] and H. Ouerdane[1]⋆

⋆ h.ouerdane@skoltech.ru

## Abstract

The bottleneck in modern thermoelectric power generation and cooling is the low energy conversion efficiency of thermoelectric materials. The detrimental effects of lattice phonons on performance can be mitigated, but achieving a high thermoelectric power factor remains a major problem because the Seebeck coefficient and electrical conductivity cannot be jointly increased. The conducting electron gas in thermoelectric materials is the actual working fluid that performs the energy conversion, so its properties determine the maximum efficiency that can theoretically be achieved. By relating the thermoelastic properties of the electronic working fluid to its transport properties (considering noninteracting electron systems), we show why the performance of conventional semiconductor materials is doomed to remain low. Analyzing the temperature dependence of the power factor theoretically in 2D systems and experimentally in a thin film, we find that in the fluctuation regimes of an electronic phase transition, the thermoelectric power factor can significantly increase owing to the increased compressibility of the electron gas. We also calculate the ideal thermoelectric conversion efficiency in noninteracting electron systems across a wide temperature range neglecting phonon effects and dissipative coupling to the heat source and sink. Our results show that driving the electronic system to the vicinity of a phase transition can indeed be an innovative route to strong efficiency enhancement, but at the cost of an extremely narrow temperature range for the use of such materials, which in turn precludes potential development for the desired wide range of thermoelectric energy conversion applications.

**1** Center for Digital Engineering, Skolkovo Institute of Science and Technology,
30 Bolshoy Boulevard, bld. 1, 121205 Moscow, Russia
**2** Center for Nonlinear and Complex Systems, Dipartimento di Scienza e Alta Tecnologia,
Università degli Studi dell'Insubria, via Valleggio 11, 22100 Como, Italy
**3** Istituto Nazionale di Fisica Nucleare, Sezione di Milano, via Celoria 16, 20133 Milano, Italy
**4** Faculty of Engineering and CENIDE, University of Duisburg-Essen, Duisburg, Germany
**5** Leibniz-IFW Dresden, Institute for Metallic Materials, 01069 Dresden, Germany
**6** Department of Physics and CENIDE, University of Duisburg-Essen, Duisburg, Germany
**7** Laboratoire Interdisciplinaire des Energies de Demain(LIED), CNRS UMR 8236,
Université de Paris, 5 Rue Thomas Mann, F-75013 Paris, France
**8** Center for Nanoscience and Nanotechnology (C2N), CNRS,
Université Paris-Saclay, 91120 Palaiseau, France

**9** Institute for Experimental Physics IV, Ruhr-Universität Bochum, Bochum, Germany

**10** NEST, Istituto Nanoscienze-CNR, I-56126 Pisa, Italy

**11** Department of Physics, Experimental Physics, Bielefeld University, Bielefeld, Germany

## Contents

## 1 Introduction

The growth of the semiconductor industry more than 60 years ago enabled significant advances in many solid-state device applications, including lasers, photovoltaics, signal amplification, and thermoelectricity [1]. These applications are now used commercially, but not all to the same extent. A comparison between thermoelectricity and photovoltaics shows the following: Their exploration began in the first half of the 19th century; both technologies benefited from the development of semiconductor physics and technology over several decades; and although

both can be considered equivalent in terms of maturity, thermoelectricity is not as widely used and integrated into power systems as photovoltaics [2,3]. Thermoelectricity is often seen as a promising solution for power generation using the vast waste heat reservoir, but so far it has not proven to be efficient or scalable to fulfill this promise [4].

Both thermoelectric and photovoltaic technologies rely on the ability of electrons to perform energy conversion [5,6], but the latter proves to be much more efficient than the former. From a thermodynamic perspective, one explanation for this difference lies in the nature of the boundary conditions to which the electronic working fluid is subjected to produce work. If energy from an external source is supplied to the electron system at temperature $T$, the change in internal energy is $\Delta U = W + Q = \mathcal{V}\Delta q + T\Delta S$, where $\mathcal{V}$ is the electric potential difference across the system, $\Delta q$ is the change in electric charge, and $\Delta S$ is the entropy change of the system. Since thermoelectric generators require a fixed temperature difference (thermal potential) to operate, while photovoltaic converters operate under a photonic flux and isothermal conditions, the electron gas response is necessarily different.

In photovoltaic converters, most of the high-grade photon energy is used to activate the valence band electrons into conduction states, since the photon energy coincides with the energy gap, and comparatively little is used for heating. In thermoelectric converters, much of the low-grade thermal energy supplied to the working fluid is dissipated to the individual degrees of freedom of the electron gas, heating it, while little is used for the collective response, i.e., electronic convective heat transport [7]. Therefore, the efficiency is low. One way to increase thermoelectric conversion efficiency is to subject the electron gas to suitable working conditions to enhance thermoelectric coupling, either by band structure engineering [8] or by preparation in a particular thermodynamic state such as a phase transition, either structurally [9–14], or electronically, as in [15, 16]. In the electronic phase transition, the focus is on the locus of energy conversion - the electronic working medium. In this way, more input energy is allocated to the collective response and the fraction of entropy that can be reversibly transported during the conversion process is maximized.

In thermoelectrics, performance is evaluated by the dimensionless ratio $zT$, which includes only the linear transport properties of materials at temperature $T$ [17]:

$$zT = \frac{\sigma\alpha^2}{\kappa}\ T = \frac{\alpha^2}{L(1 + \kappa_{\mathrm{lat}}/\kappa_{\mathrm{e}})}, \tag{1}$$

where $\alpha$ denotes the Seebeck coefficient, $\sigma$ the electrical conductivity, and $\kappa$ the thermal conductivity. Since both the electron gas and the lattice conduct heat, the total thermal conductivity $\kappa$ is $\kappa = \kappa_{\mathrm{e}} + \kappa_{\mathrm{lat}}$; and $L = \kappa_{\mathrm{e}}/\sigma T$ is the Lorenz number. Ideally, one aims to minimize heat transport by conduction, i.e., both $\kappa_{\mathrm{lat}}$ and $\kappa_{\mathrm{e}}$, while maximizing the Peltier term associated with heat transport by convection [7]. This useful heat transport mode can be characterized by the conductivity at zero electrochemical potential gradient $\kappa_{\mathrm{conv}}$, which is related to $\kappa_{\mathrm{e}}$ as follows: $\kappa_{\mathrm{conv}} = (1 + zT)\kappa_{\mathrm{e}}$ [18]. While efforts are made to increase $zT$ [19] or to optimize working conditions [20], the question of whether there is an upper bound on $zT$ is usually neglected. High values of $zT$, e.g., $zT = 4$, were considered difficult to achieve more than 20 years ago [21], and still are today. A $zT = 4$ would potentially have a maximum efficiency equivalent to about half the Carnot efficiency.[1] Only then could thermoelectric devices roughly match heat engines in terms of performance [4].

Since phonon effects cannot be completely suppressed, in this paper we look for ways to maximize the power factor. This would pave the way to a significant increase in conversion efficiency. Usually, an increase in $\alpha$ results in a decrease in $\sigma$. The reason is that the entropy per charge carrier can normally increase only by decreasing the concentration of conduction

---

[1]Taking $\eta_{\mathrm{max}} = \eta_{\mathrm{C}}(\sqrt{1+ZT}-1)/(\sqrt{1+ZT}+T_{\mathrm{cold}}/T_{\mathrm{hot}})$ as the maximal thermoelectric efficiency, and assuming $ZT = 4$ and a large temperature gap $T_{\mathrm{cold}}/T_{\mathrm{hot}} \ll 1$ to maximize the efficiency, we get $\eta_{\mathrm{max}} = 0.55\eta_{\mathrm{C}}$.

electrons; therefore, the maximum power factor is reached at charge carrier concentrations of heavily doped semiconductors [23]. Interestingly, it has been shown theoretically [24] and experimentally [25] that in quantum wells and in thin films, the Seebeck coefficient can increase without decreasing the electrical conductivity if the well width or film thickness can be reduced to dimensions smaller than the electron de Broglie wavelength. In addition, several works have shown that in strongly correlated materials, the Wiedemann-Franz law does not constrain the interrelation of Seebeck coefficients and electrical conductivity as much as in normal degenerate semiconductors and metals. In single crystal oxides [26, 27] and Kondo lattice materials [28, 29], the strong electron-electron interactions as well as the spin and orbital degrees of freedom have been found to promote improved thermoelectric properties.

In addition to strong correlations, other phenomena such as thermally induced phase transitions in electron systems can influence the thermoelectric transport parameters. Experiments with different families of superconducting compounds shows that the temperature dependence of the Seebeck coefficient has a maximum near the critical temperature after a significant increase, while the electrical resistance decreases [30–32]. Depending on the material, the change can be either abrupt or gradual. The detailed mechanisms underlying phase transitions in high-temperature superconductors are not yet fully understood. Therefore, modeling of experimental results relies heavily on phenomenological approaches to account for the peculiarities of the nonconventional superconducting materials, which include cuprates and pnictides. In particular, ferropnictides exhibit a rich phase diagram with different types of orders, e.g., orbital, magnetic, or structural, which in turn leads to an interplay between phases [33,34]. Of great interest is the nematic order (electronic state that breaks the lattice rotational symmetry) in pnictides, since it is still unclear whether unconventional superconductivity and nematicity are due to the same underlying microscopic mechanisms [35]. Furthermore, nematicity has been shown to significantly affect the thermopower [36].

The present work serves three aims: 1. to better understand the influence of the thermoelastic properties of the conduction electron gas on the coupled heat and charge transport properties; 2. to explore non-conventional ways to increase the power factor by tuning the working conditions; 3. to estimate the penalties that this increase in energy conversion efficiency entails.

Aim 1 highlights the thermodynamics of thermoelectricity, where we restrict ourselves to analytically tractable models. Discussion of phenomenological models, such as those used for nematic fluctuations, are beyond the scope of this work. Aim 2 focuses on phase transitions as a possible means of achieving large power factors. This is treated theoretically using an analytical two-dimensional model of fluctuating Cooper pairs, as a tractable model system for our approach. Further, we show experimental measurements of a thin superconducting film of an iron pnictide in which nematic fluctuations are characteristic of the phase transition, for which comparably tractable analytical models are currently lacking. However, this experimental model system has the advantage that the Seebeck coefficient signature associated with the phase transition is extremely strong. Aim 3 relies on thermodynamic calculations to provide a discussion of the trade-off between performance improvement and what the conditions that allow performance improvement implies for the use of efficient thermoelectric devices.

We focus on the conducting electron gas as an idealization of thermoelectric systems in the sense that we are interested in the conditions that enhance the performance of the working fluid that performs the energy conversion in a heat engine. We thus establish a link between the thermoelastic properties of the electronic working fluid and its transport properties. For the theoretical analysis, we consider two-dimensional electron systems (electron gas and fluctuating Cooper pairs). Our numerical results show that the closer to the superconducting phase transition, the larger the power factor can be when $\alpha$ increases, while $\sigma$ does not decrease due to the transport properties of the 2D fluctuating Cooper pairs.

In this work, we also show experimental results obtained on a 100-nm $Ba(Fe_{1-x}Co_x)_2As_2$ thin film. This experimental model system is characterized by a large nematic susceptibility related to strong nematic fluctuations. The nematic phase transition has been shown to be electronically driven [37]; it generates strong (nematic) fluctuations in the system of electrons as well as strong resistivity anisotropy [38]. This phase transition is therefore ideally suited to experimentally demonstrate the effects of electronic fluctuations on the Seebeck coefficient. We discuss our basic results from our theoretical study in light of these findings on nematic fluctuations in pnictide superconductors.

To complement our analysis, we calculate the maximum theoretical thermoelectric conversion efficiency of various electronic model systems: ideal 0-, 1-, 2- and 3-dimensional electron gases, and 2D fluctuating Cooper pairs. It is found that the efficiency goes into saturation rather quickly for all systems. The only exception is actually systems that undergo a purely electronic phase transition.

Details of the experimental data we use for the discussion, as well as formulas and calculations not shown in the main text, are included in a series of appendices.

## 2 Theory

Conduction electrons form a working fluid which has thermoelastic properties [16,39]. A dimensionless thermodynamic figure of merit $Z_{th}T$, which is directly related to the isentropic expansion factor, has been introduced as a combination of thermoelastic coefficients in Ref. [16]. The quantity $Z_{th}T$ is a measure of the energy conversion capability of the electronic working fluid from the thermostatics viewpoint; therefore, it should not be confused with the thermoelectric figure of merit $zT$ determined by the transport coefficients as in Eq. (1). In the following, using a Carnot-type approach, we focus only on the working fluid assuming everything else ideal. This means that we ignore all other sources of dissipation that negatively affect performance, such as heat leaks across the lattice and coupling with the reservoirs. While phonon effects must be taken into account in the calculations of $zT$ and the overall energy conversion efficiency, we disregard them in our thermodynamic analysis of the electronic working fluid; here we want to see what maximum efficiency the electron gas can theoretically achieve and how it compares to the Carnot efficiency of ideal heat engines. So, as the heat transfer by conduction is reduced to the electronic contribution only, $\kappa \equiv \kappa_e$ (and $\kappa \equiv \kappa_{cp}$ for the 2D fluctuating Cooper pairs – 2D FCP; see further below), the figure of merit of the electronic working fluid alone reads:

$$z_e T = \frac{\sigma \alpha^2}{\kappa_e} T = \frac{\alpha^2}{L} T, \tag{2}$$

and of course $z_e T > z T$. Formulas of the transport coefficients used to compute $z_e T$, are given in Appendix A.

### 2.1 Fundamental relations

From the thermodynamic point of view, the conduction electron gas, which transports both electric charge and energy, can be characterized by three extensive variables: internal energy $U$, entropy $S$ and charge carrier number $N$. The relationship between these variables is given by the definition of the internal energy of the system:

$$U = TS + \mu N = \left(\frac{\partial U}{\partial S}\right)_N S + \left(\frac{\partial U}{\partial N}\right)_S N, \tag{3}$$

where the intensive conjugate variables of $S$ and $N$, are the temperature $T$ and electrochemical

potential $\mu$. For an infinitesimal transformation between two equilibrium states, the fundamental relation (3) assumes the following differential form:

$$\mathrm{d}U = T\mathrm{d}S + \mu\mathrm{d}N\,, \tag{4}$$

thus yielding:

$$S\mathrm{d}T + N\mathrm{d}\mu = 0\,, \tag{5}$$

The sequence (3) $\longrightarrow$ (4) $\longrightarrow$ (5) is important: from the assumption of extensivity of the internal energy (3) comes the Gibbs relation (4), followed by the Gibbs-Duhem relation (5), which shows how heat and electricity are coupled via the intensive variables $\mu$ and $T$, and thus how thermoelectricity is deeply rooted in thermodynamics: A temperature gradient across the electron system generates a variation of its electrochemical potential with a proportionality coefficient equal to the entropy per carrier [40], $S/N = -\mathrm{d}\mu/\mathrm{d}T$, which in turn produces an electromotive force. If the temperature gradient is maintained, the nonequilibrium system reaches a steady state; if not, the system experiences transient dynamics and relaxes towards a new equilibrium state.

## 2.2 Thermoelastic coefficients and thermoelectric coupling

### 2.2.1 Definitions

The following set of equations summarizes the definitions that can be found in [16, 41]. In analogy with the classical gas, using the correspondence: $V \longrightarrow N$ and $-P \longrightarrow \mu$, we define the thermoelastic coefficients of a system of electrically charged particles:

$$\beta = \frac{1}{N}\left(\frac{\partial N}{\partial T}\right)_\mu\,, \tag{6}$$

analogue to thermal dilatation coefficient,

$$\chi_T = \frac{1}{N}\left(\frac{\partial N}{\partial \mu}\right)_T\,, \tag{7}$$

analogue to isothermal compressibility,

$$\chi_S = \frac{1}{N}\left(\frac{\partial N}{\partial \mu}\right)_S\,, \tag{8}$$

analogue to isentropic compressibility,

$$C_\mu = T\left(\frac{\partial S}{\partial T}\right)_\mu\,, \tag{9}$$

analogue to specific heat at constant pressure,

$$C_N = T\left(\frac{\partial S}{\partial T}\right)_N\,, \tag{10}$$

analogue to specific heat at constant volume.

Physically, the isothermal compressibility $\chi_T$, which is a measure of the variation of the system's volume as the applied pressure changes, can be viewed here as a capacitance in circuit theory: with the correspondence $V \to N$ and $-P \to \mu$, we see that $\chi_T$ provides a measure of the ability of the system to store electric charges under an applied voltage, so that $q^2\chi_T$

is a capacitance (in F). The application of Maxwell's relations yields the following connection between the thermal dilatation coefficient $\beta$ and the isothermal compressibility $\chi_T$:

$$\beta = \frac{1}{N}\left(\frac{\partial N}{\partial \mu}\right)_T \left(\frac{\partial \mu}{\partial T}\right)_N = \chi_T \left(\frac{\partial S}{\partial N}\right)_T , \tag{11}$$

where the notion of entropy per particle $\left(\frac{\partial S}{\partial N}\right)_T$ appears clearly as the ratio $\beta/\chi_T$. One may also see how it relates to the coupling of $\mu$ and $T$ and how a thermostatic definition of thermoelectric coupling, $\alpha_{\text{th}}$, may hence be given:

$$\alpha_{\text{th}} = \frac{1}{q}\beta \chi_T^{-1} , \tag{12}$$

as a measure of the average capacity per charged particle to transport both its electric charge $q$ and a share of the thermal energy $TS$. In this work, $q = -e$ for an electron and $q = -2e$ for a 2D fluctuating Cooper pair, with $e$ being the elementary charge.

### 2.2.2 Conduction electron gas

The thermoelastic coefficients of the noninteracting electron gas can be computed as follows [16,41]:

$$\chi_T N = \int_0^\infty g(E)\left(-\frac{\partial f}{\partial E}\right)dE , \tag{13}$$

$$\beta N = \frac{1}{T}\int_0^\infty g(E)(E-\mu)\left(-\frac{\partial f}{\partial E}\right)dE , \tag{14}$$

$$C_\mu = \frac{1}{T}\int_0^\infty g(E)(E-\mu)^2\left(-\frac{\partial f}{\partial E}\right)dE , \tag{15}$$

where $f$ is the Fermi-Dirac energy distribution function and $g$ the system's density of state. The coefficient $C_N$ can be deduced from the previous three as shown with the calculation of the isentropic expansion factor shown further below. Note that unlike the transport coefficients (given in Appendix A) these thermoelastic coefficients depend on $g(E)$ and $f(E)$ but not on the transport distribution function, which involves the electron speed and the relaxation time.

### 2.2.3 Fluctuating Cooper pairs

Fluctuating Cooper pairs, are bosonic quasi-particles that exist in the metal phase when the pairing mechanism that binds two conduction electrons is maintained above the critical temperature of the superconducting phase transition $T_c$ [42]. A fluctuating Cooper pair typical size can reach the $10^3$ to $10^4$ Å range, while their size is in a more limited range in iron-based systems such as Co-doped $BaFe_2As_2$ single crystals, i.e. 10 to 100 Å [43]. While in bulk clean metals pairing due to thermal fluctuations above $T_c$ is possible only over a very small temperature range, pairing can occur well above $T_c$ in thin films as dimensionality and disorder also play a role in the pairing mechanism [44,45]. The analysis of the working fluid constituted of 2D FCP thus necessitates a different approach as they do not form a non-interacting Fermi gas. In the theoretical part of the present work, we focus on the effect of 2D fluctuating Cooper pairs on the thermoelectric power factor close to $T_c$.

The chemical potential $\mu_{cp}$ of a system with $N_{cp}$ pairs is derived from the free energy $\mathcal{F}_{cp}$ [42]:

$$\mathcal{F}_{cp} = -\frac{A}{4\pi\xi^2} k_B T_c \varepsilon \ln \varepsilon \,, \tag{16}$$

$$\mu_{cp} = \frac{\mathcal{F}_{cp}}{\partial \varepsilon} \times \left(\frac{\partial N_{cp}}{\partial \varepsilon}\right)^{-1} \,, \tag{17}$$

where $\varepsilon = \ln T/T_c \approx (T - T_c)/T_c$, $\xi$ is the coherence length, $k_B$ the Boltzmann constant, and $A$ is the surface area of the 2D system. From Eqs. (16) and (17), we can obtain the following quantities:

The thermostatic definition of the 2D FCP thermoelectric coupling [46]:

$$\alpha_{th,cp} = q^{-1}\frac{d\mu_{cp}}{dT} \,. \tag{18}$$

The coefficient $C_N$:

$$C_{N_{cp}} = -T\frac{\partial^2 \mathcal{F}_{cp}}{\partial T^2} \,. \tag{19}$$

And the coefficient $\chi_{T_{cp}}$:

$$\chi_{T_{cp}} = \frac{1}{N_{cp}}\left(\frac{\partial N_{cp}}{\partial \mu_{cp}}\right)_T \,. \tag{20}$$

The coefficient $C_{\mu_{cp}}$ can be deduced from the previous three as shown with the calculation of the isentropic expansion factor shown further below.

## 2.3 Isentropic expansion factor and thermodynamic figure of merit

In classical thermodynamics, the isentropic expansion factor $C_P/C_V$ is a measure of the ability of a working fluid to convert heat into work. The larger $C_P$ is in relation to $C_V$, the more heat is converted into mechanical work at constant pressure: The working fluid expands when it receives thermal energy, whereas it cannot expand at constant volume, in which case the thermal energy only heats up the system. The correspondence between the conjugate thermodynamic variables of a classical working fluid, the volume $V$ and the pressure $-P$, and the variables relevant to a conduction electron gas, the number of electrons $N$ and the electrochemical potential $\mu$: $V \longrightarrow N$ and $-P \longrightarrow \mu$, gives the thermoelectric heat capacity ratio $\gamma$ at temperature $T$ [16, 39]:

$$\gamma = \frac{C_\mu}{C_N} = 1 + \frac{\beta^2}{\chi_T C_N}T = 1 + \frac{\alpha_{th}^2}{\ell} = 1 + Z_{th}T \,. \tag{21}$$

Equation (21) is the definition of the thermodynamic figure of merit $Z_{th}T$ of the electronic working fluid, which has a formal similarity with $z_e T$ in Eq. (2). The quantity $\ell = C_N/q^2\chi_T T$ is the thermostatic counterpart of Lorenz number $L$ in coupled transport; while the latter measures the ability of the system to conduct thermal energy relative to its ability to conduct electricity, the former measures the ability of the system to store thermal energy relative to its ability to gain conduction electrons [16]. Thus, in the context of thermoelectricity, $\ell$ should be small so that the electron gas tends to minimize heat transfer by conduction, while $\alpha_{th}$ should be high so that the Peltier contribution to heat flow or heat transfer by convection [7, 16], i.e., electric charge transport, is maximized. Furthermore, if conditions are found where $C_\mu/C_N$ reaches high values or even diverges (such as near a phase transition [16, 39]), the system would tend to behave like an ideal working fluid with high efficiency in converting

heat into work. In this work, we numerically calculate the isentropic expansion factor $\gamma$ for the noninteracting electron gases (0D, 1D, 2D, and 3D) using the formulas (13) – (15).

The same analysis applies to the case of a fluctuating 2D Cooper pair gas. Using Eqs. (18)– (20) an analytical formula can be established: $\gamma_{cp} = 1 - \ln \varepsilon$, and an expression of the thermodynamic figure of merit of the 2D FCP gas follows:

$$Z_{\text{th,cp}}T = \gamma_{cp} - 1 = -\ln \varepsilon \,. \tag{22}$$

### 2.4 Transport coefficients near the superconducting phase transition

We now turn to the 2DEG just above the critical temperature $T_c$ and its thermoelectric properties driven by the 2D fluctuating Cooper pairs [42]. The existence of fluctuating Cooper pairs above $T_c$ gives rise to paraconductivity, which is a pair contribution that enhances the electrical conductivity $\sigma_{cp} = e^2/16\hbar\varepsilon$ [42,47]. The thermal conductivity of fluctuating Cooper pairs is $\kappa_{cp} = (k_B^2 \alpha_{GL}^2 T_c/64\hbar)\varepsilon \ln(1/\varepsilon)$, with $\alpha_{GL} = 4\pi^2/[7\zeta(3)]k_B T_c/E_F^{2D}$ being a dimensionless parameter in the Ginzburg-Landau free energy functional [42,48], and $E_F^{2D}$ the 2D Fermi energy. The Seebeck coefficient being given by $\alpha_{cp} = \nabla\mu/q\nabla T = \alpha_{GL}k_B \ln \varepsilon/2e$ [48,49], the thermoelectric figure of merit of the 2D FCP in the fluctuating regime near $T_c$ can be expressed as:

$$z_{cp}T = \frac{1}{\varepsilon^2} \ln \frac{1}{\varepsilon} \,, \tag{23}$$

which diverges as the electron system gets close to the critical temperature, i.e., as $\varepsilon \longrightarrow 0$. Note that the difference between the figure of merit $z_{cp}T$ defined using the transport coefficients and the thermodynamic figure of merit of the 2D FCP working fluid $Z_{\text{th,cp}}T$ defined using the thermoelastic coefficients, is simply in their temperature dependence.

It is interesting to note that in the limit $T \to T_c$, $\kappa_{cp} \to 0$ while $\kappa_{\text{lat}}$ may remain finite, but this does not pose any problem in terms of performance of the ideal 2D FCP system, as the system tends to become an "electron crystal" and this fact contributes to make $z_{cp}T$ larger. Including $\kappa_{\text{lat}}$ in the calculation of full figure of merit $zT$ would negatively impact on the performance, but here we see that without heat leaks, the conversion efficiency can tend to the Carnot efficiency of an ideal heat engine in the limit $T \to T_c$.

It is also instructive to calculate the Lorenz number for the 2D fluctuating Cooper pairs $L_{cp} = \kappa_{cp}/(\sigma_{cp}T)$:

$$L_{cp} = \frac{\alpha_{GL}^2}{4}\left(\frac{k_B}{e}\right)^2 \times \frac{(T - T_c)^2}{T T_c} \ln\left(\frac{T_c}{T - T_c}\right). \tag{24}$$

As the system's temperature tends to $T_c$, $L_{cp} \to 0$, which clearly deviates from the standard Lorenz number: $L = \kappa/(\sigma T) = \pi^2/3(k_B/e)^2$. In this case, the power factor $\sigma_{cp}\alpha_{cp}^2 \propto (\ln \varepsilon)^2/\varepsilon$ shows a diverging behavior in the limit $\varepsilon \longrightarrow 0$. The thermoelectric working fluid in the fluctuating region near $T_c$ thus acquires the desired properties for higher conversion efficiency.

## 3 Results

### 3.1 Correlations between thermoelectric figure of merit and thermodynamic figure of merit

Ideally, thermoelectric conversion should be isentropic: all the thermal energy supplied to the conduction electron gas should serve for convective heat transport [7] and the entropy production should also be zero. The relation $\kappa_{\text{conv}} = (1 + z_e T)\kappa_e$ [18] is similar to $\gamma$ in Eq. (21) and thus defines an isentropic expansion factor in the context of electron transport: $\gamma_{\text{tr}} = \kappa_{\text{conv}}/\kappa_e = 1 + z_e T$. In this way, the thermodynamic figure of merit $Z_{\text{th}}T$ of the conduction

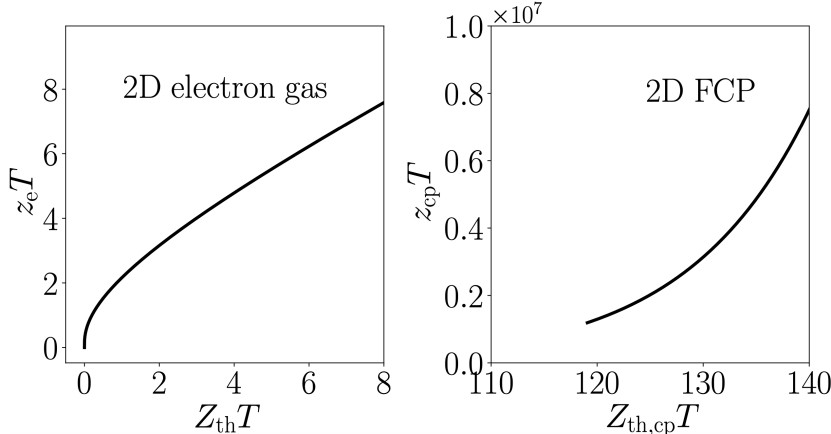

Figure 1: Parametric plots of $z_e T$ vs $Z_{th} T$ of the 2D electron gas (with $\kappa \equiv \kappa_e$) on the left panel, and $z_{cp} T$ vs $Z_{th,cp} T$ of the 2D fluctuating Cooper pairs (with $\kappa \equiv \kappa_{cp}$) on the right panel. The lowest value here taken by $Z_{th,cp} T$ corresponds to the temperature range imposed to be in the vicinity of the superconducting phase transition and for the model to be valid. Note the ultra-high values of $z_{cp} T$ of the 2D FCP system, which stem from its extremely strong dependence on the temperature, Eq. (23). The temperature range for the 2D electron gas goes up to 300 K, while it goes from $T_c$ to $T_c + 0.2$ K, for the 2D FCP simulation. The values of the parameters used for the numerical calculations are given in Appendix A.

electron gas, which is directly related to the isentropic expansion factor $\gamma = C_\mu / C_N = 1 + Z_{th} T$, connects the thermodynamic and transport properties of the electronic working fluid at temperature $T$.

In Fig. 1, the correlations between $z_e T$ and $Z_{th} T$, and between $z_{cp} T$ and $Z_{th,cp} T$, give the deviation of $\gamma_{tr}$ from the standard $\gamma$ in thermostatics, in both the normal thermodynamic regime and the fluctuating regime in the 2D systems. The curves show a monotonic increase of $z_e T$ against $Z_{th} T$. Note the different magnitudes between the case where a phase transition occurs and the one where it does not (see also the additional curves for 0D, 1D, and 3D systems in Appendix B). This clearly shows that the power factor $\sigma \alpha^2$ can reach extremely high values if suitable boundary conditions are imposed on the electron gas. It is instructive to see how this theoretical result can be related to experiments close to $T_c$, in the vicinity of a phase transition.

## 3.2 Measured Seebeck coefficient and electrical conductivity in a Ba(Fe$_{0.90}$Co$_{0.10}$)$_2$As$_2$ thin film

The experimental data used in this study are from an epitaxially grown 100-nm thick pnictide Ba(Fe$_{0.90}$Co$_{0.10}$)$_2$As$_2$ thin film on a CaF$_2$ substrate, which exhibits extremely good structural and compositional quality. The superconducting properties of comparable samples were previously described in Ref. [50]. After characterizing the transport properties, we introduced defects into the thin film by ion bombardment with 10 keV argon ions. The same but degraded sample was then characterized again. Details are given in Appendix C.

Figure 2 shows the Seebeck coefficient and electrical resistivity of the thin film. The measurements shown in the upper panel were performed on a sample with high structural quality, while the lower panel shows data from the same sample with low structural quality after ion bombardment. The Seebeck coefficient clearly shows the onset of nematic fluctuations. For a sample of this composition, one would expect the strong enhancement of nematic fluctuations below $\sim$50 K [38, 51]. It is obvious that these fluctuations have little effect on the electrical

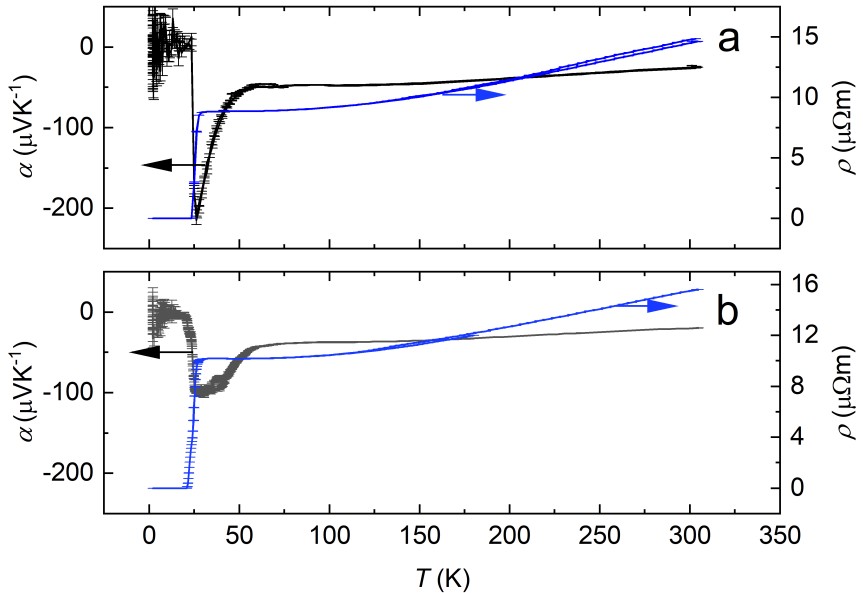

Figure 2: Thermoelectric transport characterization of the 100-nm $Ba(Fe_{0.90}Co_{0.10})_2As_2$ thin film showing the Seebeck coefficient $\alpha$ and electrical resistivity $\rho$: a) high structural quality before ion bombardment, and b) low structural quality after ion bombardment.

resistance. The transition from an electron system without fluctuations to an electron system with fluctuations is not evident in $\rho$. However, this can be understood qualitatively by considering the influence of a fluctuation regime near a phase transition on the Seebeck coefficient. Small fluctuations that are not noticeable in resistivity can already have significant effects on the Seebeck coefficient. In other words, from the thermodynamic viewpoint, electronic fluctuations modify the thermostatic properties of the electron gas in such a way that the thermoelectric coupling or entropy per electron increases, while the electrical conductivity is not as much influenced. Note that the traditional descriptions of the Seebeck coefficient, e.g., in the Boltzmann model, which often argue with the change in conductivity, cannot help in interpreting the data here.

Structural quality has a limited effect on both the Seebeck coefficient and electrical resistivity away from the superconducting phase transition region and above the onset for nematic fluctuations, here above 50 K. At lower temperatures, however, it has a strong effect on the Seebeck coefficient, as shown by the dramatic reduction in the magnitude of $\alpha$, which contrasts strongly with the temperature dependence shown in the upper panel. While defects strongly impact superconducting fluctuations, this remains unclear when it comes to nematic fluctuations. We note however, that the nematic transition temperature does not change dramatically with irradiation. The physical interpretation of these observations necessitate a model of the influence of structural defects and disorder on the thermostatics of the electron gas in the fluctuation regime, which is beyond the scope of the present work.

In Fig. 3, the power factor (normalized to its value at $T = 300$ K) calculated using the transport coefficients of the 2DEG and 2D FCP is shown and compared with experimental data obtained with the high and then low structure quality samples (after ion bombardment). These 2D models are of course highly idealized, however the 2D FCP, though based on restrictive assumptions, may support the idea that a strongly fluctuating regime in the close vicinity of a critical point fosters the enhancement of the Seebeck coefficient. More realistic numerical data would at least require, for example, a density of states modified by a thickness-dependent form factor and electronic correlations to be taken into account.

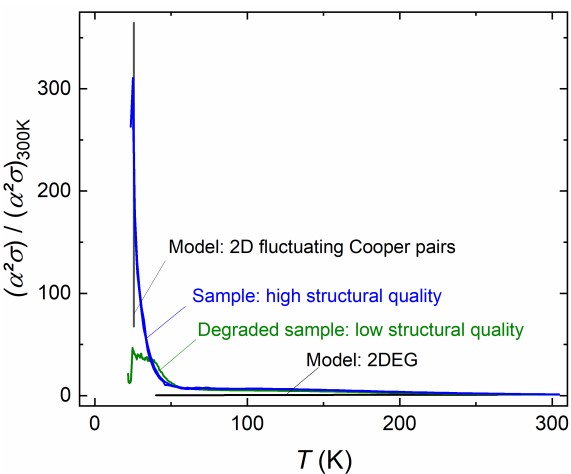

Figure 3: Power factors $\sigma\alpha^2$ normalized to their value at 300 K, of the 100-nm thick $Ba(Fe_{0.90}Co_{0.10})_2As_2$ thin film before and after the ion bombardment, as functions of temperature. The transition region between the normal regime and the superconducting phase is influenced by the structural quality of the sample. Note the steep, nearly vertical slope computed for the ideal 2D FCP system, near $T_c = 25.6$ K, as $\sigma_{cp}\alpha_{cp}^2$ is proportional to $\ln^2(\varepsilon)/\varepsilon$, with $\varepsilon = (T - T_c)/T_c$. The 2DEG model, which is an approximation for a thin film, appears to be in relatively good qualitative agreement only in the high temperature regime.

## 4 Discussion

### 4.1 Experimental data interpretation

An increase in the absolute value of the Seebeck coefficient before entering the superconducting region has been reported in the literature. Fairly sharp peaks of the Seebeck coefficient, comparable to our experimental results, were measured in nearly ideal electron-doped $La_{2-x}Ce_xCuO_4$ thin films. Both sharp and broadened Seebeck peaks were detected for $SmFeAsO_{1-x}F_x$ and $Fe_{1+y}Te_{1-x}Se_x$ [32]. In [30], a strong increase in the Seebeck coefficient of a series of pnictides was shown in which the Sm lattice position was replaced by Nd and La. This increase was accompanied by a strong improvement in the power factor. Broadened peaks of the Seebeck coefficient were found in different pnictide compositions [52,53] as well as in cuprates [54].

A steep increase in the Seebeck coefficient before entering the superconducting state is also known for elementary superconductors, e.g., Pb and Nb superconductors [55,56]. This effect is traditionally explained by the phonon resistance effect [57,58]. In this study, we have shown that even in the absence of phonons, a sharp increase in the Seebeck coefficient would be possible, which would have its origin in fluctuations of the electron system. Our models focus only on the thermodynamics of the ideal electronic systems, and as shown in Fig. 3, they do not take into account the structural quality of the thin films, which affects the behavior of the Seebeck coefficient when the temperature varies and leads the system from the normal regime to the fluctuating regime.

Modeling our experimental data beyond the superconducting fluctuating region is a task that would require a full separate work. Detailed knowledge of the band structure and phonon spectrum, taking into account the effects at the film/substrate interface, would then be required. The experimental data show that the dependence of the Seebeck coefficient on temperature after degradation of the sample by ion bombardment does not vary greatly as the

temperature drops because of other processes such as the scattering of phonons by defects that suppress phonon drag effects, on the one hand, and the defects that prevent fluctuations from occurring, on the other hand. A study of the phonon drag effect in thin films in the normal regime shows that the Debye temperature of the substrate influences the position and magnitude of the phonon drag peak as the drag effect varies with temperature [59]. Interestingly, it was also shown that the phonon drag effect is strongly suppressed with film thickness. Since the film thickness in our sample is 100 nm, we can assume that the phonon drag effect plays a role but is not dominant near $T_c$. This is acceptable for the analysis of the behavior of the Seebeck coefficient and the power factor near the critical temperature $T_c$.

While this was not studied in further detail within the scope of this work, we suggest that the variation of the peak shape of the Seebeck coefficient, i.e. sharp or broadened, is a result of structural or compositional inhomogeneities in these samples. With a quite good structural and compositional integrity of our sample, the data here presented serves as a model system to underline the influence of fluctuations. Close to the critical temperature, the Seebeck coefficient and the electrical conductivity evolve differently; Eq. (23) shows clearly the resulting temperature dependence of $z_{cp}T$ close to the superconducting phase transition. This illustrates the difference between the transport of charges characterized by $\sigma$, which is proportional to the carrier concentration, and the transport of entropy characterized by $\alpha$, which varies logarithmically with the concentration. Being close to $T_c$ favors a more rapid variation of $\alpha$. That said, more work is required to develop a model dedicated to the influence of nematic fluctuations on the transport coefficients to adequately describe the observed rise of the power factor from around 50 K as the temperature decreases. We suggest here that the clear and distinct increase of the Seebeck coefficient observed in our experiment is a consequence of driving the subsystem of conduction electrons to fluctuating regimes.

## 4.2 Ideal maximum efficiency

Turning to the maximum thermoelectric conversion efficiency that the electronic working fluid can boast, we show two approaches for its evaluation. First, the standard formula [18], which here we adapt considering only the electronic working fluid subjected to the temperature gradient $T_{hot} - T_{cold}$ when a thermoelectric generator is in (ideal) thermal contact with a heat source at temperature $T_{hot}$, and a heat sink at temperature $T_{cold}$:

$$\eta_{max}^{tr} = \frac{\sqrt{\gamma_{tr}} - 1}{\sqrt{\gamma_{tr}} + T_{cold}/T_{hot}} \eta_C = \frac{\sqrt{1 + z_e T} - 1}{\sqrt{1 + z_e T} + T_{cold}/T_{hot}} \eta_C, \tag{25}$$

where $z_e T$ is defined in Eq. (2) and is related to $\gamma_{tr}$ introduced in Section 3, $\eta_C = 1 - T_{cold}/T_{hot}$ is the Carnot efficiency, and the superscript tr stands for transport, as here this efficiency is evaluated considering the transport coefficients. Since in this work, we are also interested in the thermoelastic properties of the electronic working fluid, another expression for the maximum thermoelectric efficiency based on these properties can be derived, and $\eta_{max}$ is related to the heat capacity ratio $\gamma$ as follows [60, 61]:

$$\eta_{max}^{th} = \frac{\sqrt{\gamma} - 1}{\sqrt{\gamma} + 1} \eta_C = \frac{\sqrt{1 + Z_{th} T} - 1}{\sqrt{1 + Z_{th} T} + 1} \eta_C, \tag{26}$$

where $Z_{th} T$ is defined in Eq. (21). Fomrulas for the 2D FCP case are formally the same with the following substitutions: $z_e T \rightarrow z_{cp} T$, and $Z_{th} T \rightarrow Z_{th,cp} T$.

Note the formal similarity between $\eta_{max}^{tr}$ and $\eta_{max}^{th}$ as expressed in Eqs. (25) and (26). The former evaluates the performance during the coupled transport of charge and electricity under a temperature bias, while the latter evaluates the "quality" of the electronic working

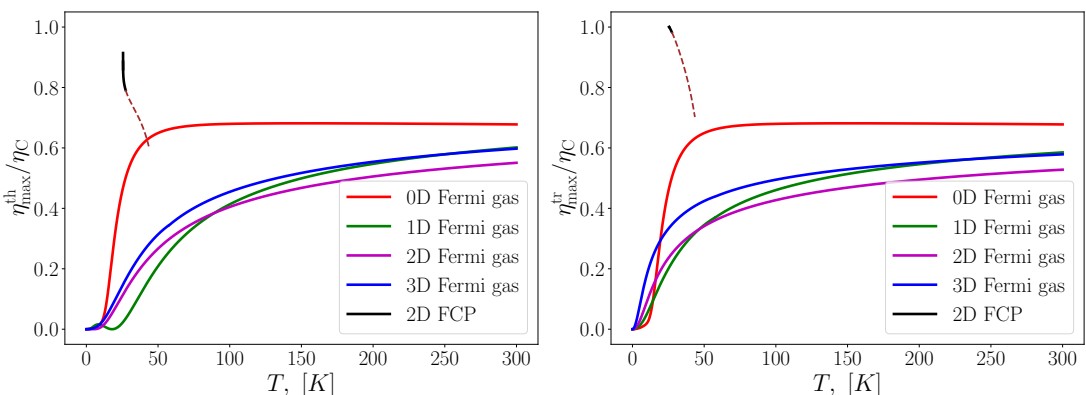

Figure 4: Maximum thermoelectric conversion efficiencies $\eta_{max}^{th}$ (left panel) and $\eta_{max}^{tr}$ (right panel) attainable by the electronic working fluids considered in this work, as functions of temperature. The ratio $T_{cold}/T_{hot}$ in Eq. (25) is kept at a constant value of 0.95 for the numerical calculation of $\eta_{max}^{tr}$. The dashed lines on both panel show when the 2D FCP model becomes invalid.

fluid at temperature $T$. In Eq. (25), $T$ is the average temperature across the system and we simply assume $T = (T_{hot} + T_{cold})/2$, while in Eq. (26), $T$ is the equilibrium temperature, which amounts to taking $T_{hot} = T_{cold}$, and hence a ratio of these temperatures equal to 1. Owing to the relationship between $Z_{th}T$ and $z_e T$, Eq. (26) confirms the well-known result that for a given set of boundary conditions, the larger $zT$ the larger $\eta_{max}$.

The maximum efficiencies $\eta_{max}^{tr}$ and $\eta_{max}^{th}$ scaled to the Carnot efficiency calculated for all model systems considered in this work, are reported in Fig. 4. The curves demonstrate that though ideal cases are considered (the focus being on the working fluid only, and a situation with no dissipative coupling between the generator and the thermal energy reservoirs), the Carnot efficiency cannot be reached; for this to occur, the isentropic expansion factor of the conduction gas would have to diverge, implying that it would have the capability to convert almost all of the thermal energy it receives into work, or convective heat transport [7]. We saw in light of our thermodynamic analysis and experimental data, that bringing the electron gas close to a phase transition enhances the system's power factor $\sigma\alpha^2$; hence by increasing both the thermoelectric coupling and electrical conductivity, thus favoring convective transport, the conversion of heat into work can be significantly boosted, albeit in a very limited temperature range in our study. Figure 4 also shows that while increasing the temperature of the electron systems that do not undergo a phase transition, yields an increase of $\eta_{max}$, their maximum efficiencies grow slowly and seem to saturate. For the system in the vicinity of a phase transition, the efficiency may tend to that of the ideal Carnot efficiency. Importantly, as we focused on the electronic working fluid only, the existence of an upper bound to $zT$ accounting for the lattice thermal conductivity is demonstrated: for each system considered, the upper bound corresponds to that shown in Fig. 4.

## 5  Conclusion

The thermoelastic properties of the electron gas in the normal phase are such that even in an ideal situation, notwithstanding detrimental phonon effects and other external causes of heat leaks and dissipation, only up to about half of the thermal energy fed to the electrons can be used for their collective response, i.e. electrical current. A path forward may lie in

finding very specific working conditions such as placing the electron gas in the vicinity of a phase transition that favorably modifies the heat capacity ratio, and in turn the electronic transport properties as shown by the substantial increase of the power factor. The theory and experiment reported in this paper describe different thermodynamic conditions but tend to independently show that efforts are to be invested in the control of the electronic transport properties close to a phase transition: the 2D fluctuating Cooper pairs close to $T_c$ and the nematic fluctuations over a wider temperature range foster the desired transport properties, resulting in a strong enhancement of the power factor. As already hinted in [16], the more compressible a working fluid is, the larger the heat capacity ratio is; so, our conclusion on the benefits that fluctuating regimes (here superconducting or nematic) can bring is consistent with the fluctuation-compressibility theorem [62, 63] in the context of thermoelectricity: the larger the electronic fluctuations are, the larger the isothermal compressibility $\chi_T$ is.

On the flip side, phase transitions impose that a thermoelectric generator operates only extremely close to the transition temperature and with a corresponding very small temperature bias. We thus see with the 2D fluctuating Cooper pairs that close to $T_c$, the conversion efficiency can theoretically tend to the Carnot efficiency, but at the cost of narrowing down the applicability of efficient thermoelectric solutions for waste heat conversion. From the thermodynamic viewpoint this amounts to stating that the minimization of entropy production during the thermoelectric energy conversion necessitates strong constraints that preclude its use for power production beyond niche applications. Depending on the degree of control of the working conditions, a high power factor obtained close to a phase transition might rather prove more useful for pumping heat.

We finally emphasize the need to develop new experiments to further explore the physics of thermoelectricity in fluctuating regimes to bring new data and stimulate more theoretical research along the lines developed here, notably to develop more realistic models based on less restrictive assumptions than those used in the present work. Unconventional superconductors and strongly correlated systems offer a challenging yet rich field of play for that purpose.

## Acknowledgments

I.K. thanks Alberto Parola for fruitful discussions.

## A  Transport coefficients in the relaxation time approximation

For our numerical calculations, we consider low-density noninteracting electron gases with concentrations $n_{3D} = 10^{18}$ cm$^{-3}$, $n_{2D} = 10^{12}$ cm$^{-2}$, and $n_{1D} = 10^{6}$ cm$^{-1}$ for the three-, two-, and one-dimensional systems, respectively. The corresponding Fermi energies are $E_F^{3D} = 3.64$ meV, $E_F^{2D} = 2.39$ meV, and $E_F^{1D} = 0.94$ meV. The concentration of 2D fluctuating Cooper pairs is $n_{cp} = 10^{12}$ cm$^{-2}$.

The transport coefficients for a variety of systems including low-temperature [64] and interacting systems [65, 66] can be calculated using the Boltzmann equation. The simplest assumption is that of the relaxation time approximation for the electrons [67, 68], in which

case Onsager's kinetic coefficients can be calculated as follows:

$$L_{11} = \frac{T}{d} \int_0^\infty \Sigma(E) \left( -\frac{\partial f}{\partial E} \right) \mathrm{d}E \,, \tag{A.1}$$

$$L_{12} = L_{21} = \frac{T}{d} \int_0^\infty (E - \mu) \, \Sigma(E) \left( -\frac{\partial f}{\partial E} \right) \mathrm{d}E \,, \tag{A.2}$$

$$L_{22} = \frac{T}{d} \int_0^\infty (E - \mu)^2 \, \Sigma(E) \left( -\frac{\partial f}{\partial E} \right) \mathrm{d}E \,, \tag{A.3}$$

where $d$ is the system's dimension, $\Sigma(E) = \tau(E) \, v^2(E) \, g(E)$ is the transport distribution function with $\tau$ being the relaxation time, $v$ the velocity, and $g$ the density of states [69]. For the chemical potentials we used analytical expressions given in Ref. [70]. Note that for the 1D and 0D cases, we used the numerical values obtained for the calculation of the 2D electrochemical potential. Note that we assume parabolic bands so $v^2(E) = 2E/m$. The relaxation time depends on various scattering processes, involving electron-electron interaction, electron-phonon interaction, ionized impurity scattering, and several others. These processes influence the electron mobility, which is given by $q\tau/m$. In the present work, we consider the electronic working fluid only, and the easiest approach for numerical calculations, is to make the additional approximation of constant relaxation time based on typical values of the electronic mobility. Choosing the values of the electron mobility as 3000 cm$^2$/V s [71] for the electron gas in a bulk system, 3450 cm$^2$/V s for the electron gas in a quantum well [72], and 1200 cm$^2$/V s [73] for the electron gas in a nanowire, we thus use the following relaxation times: $\tau_{3D} = 1.7$ ps, $\tau_{2D} = 1.95$ ps, and $\tau_{1D} = 0.68$ ps, for our numerical calculations.

The density of states of the isotropic 3D, 2D, and 1D noninteracting electron systems with parabolic energy dispersion are given by:

$$g_{3D}(E) = \frac{m}{2\pi^2\hbar^3} \sqrt{2mE} \,, \tag{A.4}$$

$$g_{2D}(E) = \frac{m}{2\pi\hbar^2} \,, \tag{A.5}$$

$$g_{1D}(E) = \frac{1}{2\pi\hbar} \sqrt{\frac{2m}{E}} \,. \tag{A.6}$$

For the 0D electron gas, we use a single-level quantum dot and a Lorentzian form for the density of states:

$$g_{0D}(E) = \frac{\Gamma}{(E - E_0)^2 + (\Gamma/2)^2} \,, \tag{A.7}$$

where $\Gamma$ is the energy level width. The central energy of the channel is $E_0 = 2.37$ meV, and the channel coupling energy $\Gamma = 0.1 k_B T$.

The transport coefficients read [18]:

$$\sigma = \frac{e^2 L_{11}}{T} \,, \tag{A.8}$$

$$\alpha = \frac{L_{12}}{q T L_{11}} \,, \tag{A.9}$$

$$\kappa_e = \frac{1}{T^2} \left[ L_{22} - \frac{L_{12} L_{21}}{L_{11}} \right] \,, \tag{A.10}$$

where $\sigma$ is the isothermal electrical conductivity, $\alpha$ is the Seebeck coefficient, and $\kappa_e$ is the thermal conductivity under zero electric current.

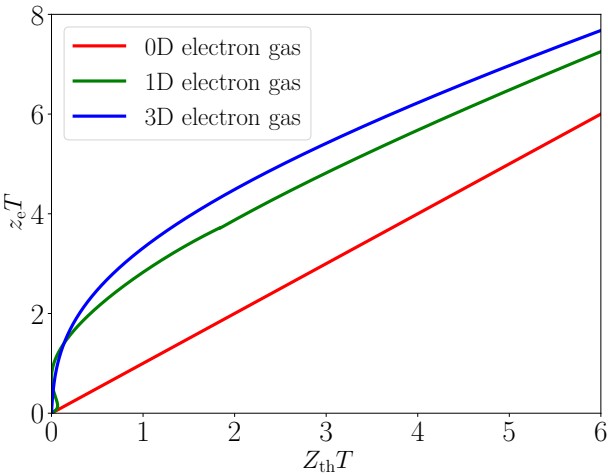

Figure 5: $z_e T$ vs $Z_{th} T$ for the 0D, 1D and 3D electron gases.

# B   Additional $z_e T$ vs. $Z_{th} T$ curves

Here we show, the curves not shown in the main text, i.e. for the 0D, 1D and 3D electron gases. The 2D and 2D FCP cases are shown in Fig. 1. The thermoelastic coefficients and the transport coefficients for all systems can be computed from the formulas given above and combined as shown in Eqs. (1) and Eq. (21) in the main text, to obtain $Z_{th} T$ and $z_e T$ (with $\kappa \equiv \kappa_e$). As discussed in the main text, the parametric plot of $z_e T$ against $Z_{th} T$ shows a clear correlation between the isentropic expansion factor and the thermoelectric figure of merit: the larger the former, the larger the latter.

All the curves depicted in Fig. 5 show a monotonic behavior. It is interesting to note that for the 0D system, the correlation is linear implying that the isentropic expansion factor in the transport regime does not deviate from the heat capacity ratio at equilibrium, while it does for the small values of $z_e T$ and $Z_{th} T$ for finite-dimensional systems. This originates in the energy dependence of the transport distribution functions and of the density of states in particular.

# C   Experimental

The $Ba(Fe_{0.90}Co_{0.10})_2As_2$ thin film composition corresponds to the close-to-ideal doping case for these epitaxial growth conditions [75], reflected by a high transition temperature $T_c$ of 25.6 K. The thin film was prepared by pulsed laser deposition method in ultra-high vacuum of $10^{-9}$ mbar utilizing a KrF excimer laser. The pnictide thin film was grown with a frequency of 7 Hz at 700 °C, while its thickness was controlled by the number of pulses. Similar pnictide thin films were studied in detail in Refs. [50, 75, 76].

The thermoelectric properties of the sample were characterized by a Physical Property Measurement System of the Quantum Design DynaCool series (9 T), equipped with a thermal transport option. Utilizing the thermal transport option, the experimental parameters Seebeck coefficient and electrical resistivity were measured simultaneously and continously as a function of temperature. Hereby, the sample was subjected to a thermal pulse, and its temperature and voltage responses were recorded. The Seebeck coefficient was extracted from these data, and the resistivity was characterized subsequently. Electrical contacts were made by a conducting silver-particle based two component epoxy glue that is recommended from

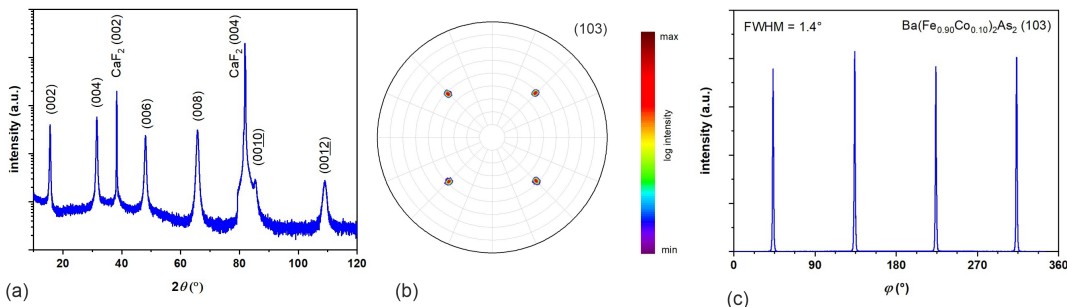

Figure 6: Structural characterization. (a) XRD pattern of the standard Ba(Fe$_{0.90}$Co$_{0.10}$)$_2$As$_2$ film using Co-K$_\alpha$ radiation; (b) pole figure for the (103) plane of Ba(Fe$_{0.90}$Co$_{0.10}$)$_2$As$_2$; (c) analysis of the in-plane alignment using a $\phi$ scan through the (103) peaks.

Quantum Design for the thermal transport option. Note that both Seebeck coefficient and electrical resistivity could only be characterized in the normally conducting state of the sample, since $\alpha = 0$ and $\rho = 0$ in the superconducting state. The thermal transport option of the DynaCool, in principle, also provides thermal conductivity data. But for this thin film sample, these data were completely dominated by the thermal conductivity of the CaF$_2$ substrate, and are therefore not shown here.

## C.1 X-ray diffraction characterization

The structure of the grown film was studied using X-ray diffraction (XRD) using a Bruker D8 Advance diffractometer with Co-K$_\alpha$ radiation for standard $\theta - 2\theta$ measurements and a Panalytical X'Pert system with Cu-K$_\alpha$ radiation for texture measurements.

XRD studies showed only (00$\ell$) peaks in the $\theta - 2\theta$ scans indicating a clear c-axis orientation of the film (Fig. 6(a)). The c-axis lattice parameter was determined to 1.319 nm. Texture measurements revealed an epitaxial growth with a sharp in-plane alignment having a full width at half maximum (FWHM) value of 1.4° (Figs. 6(b,c)). The results are almost identical to data published previously on Ba(Fe$_{0.90}$Co$_{0.10}$)$_2$As$_2$ films prepared under similar deposition conditions [50]. Therefore, we assume also a similar clean microstructure with a small reaction layer towards the substrate as shown by high resolution transmission electron microscopy in this work.

## C.2 Ion bombardment

To introduce defects into the 100-nm-thick Ba(Fe$_{0.90}$Co$_{0.10}$)$_2$As$_2$ thin film deposited on the CaF$_2$ substrate, it bombarded with argon ions with a kinetic energy of 10 keV and with a fluence of 8.5×10$^{12}$ ions per cm$^2$ hitting the sample under an angle of incidence of 45°. In this energy regime, the damage is almost exclusively due to nuclear stopping, i.e. binary collisions between atoms. The resulting collisional cascades of ions (see Fig. 7a) and recoil atoms (see Fig. 7b) lead to the creation of a defective zone with an extension on the order of typically a few ten nanometers. To estimate the extent of ion-induced damage for the samples studied here we ran model calculations with the software package "Stopping and Range of Ions in Matter" (SRIM-2008) [77] which computes the interactions of energetic ions with amorphous targets using a Monte Carlo approach. As input parameters, we used the above mentioned ion beam parameters, a density of 6.47 g·cm$^{-3}$ and the stoichiometry of the sample, in combination with generic values for the otherwise unknown lattice binding (3 eV) and displacement (25 eV) energies for all target elements, and the respective elementary surface binding energies

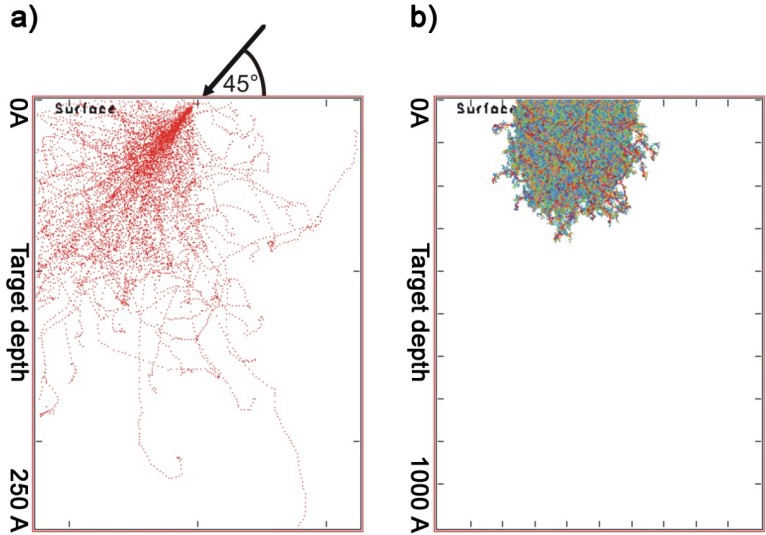

Figure 7: a) Simulated ion trajectories in a 25 nm thick $Ba_2(Fe_{0.9}Co_{0.1})_2As_2$ compound as calculated with SRIM-2008. Left: The black arrow depicts the direction of the incoming Ar ion beam. Red dots depict projectile positions until the ion is finally stopped (black dot), here for 200 projectiles. For clarity, recoil atoms have been omitted. b) The extension of the damaged zone into the thin film can be estimated by overlaying a sufficient number of simulated trajectories, here 5000. Recoil atoms are plotted as well using the following colour code: Ba green, Fe blue, Co pink, As orange.

(Ba: 1.84 eV, Fe: 4.34 eV, Co: 4.43 eV, As: 1.26 eV) provided by SRIM-2008. We used the full cascade mode, in which the collisional damage to the target is analyzed by following every recoil atom until its energy drops below the lowest displacement energy of any of the target atoms. From these calculations, one may infer the extension of the defective zone into the film. Figure 7 shows that the bombardment of the film has led to a defective zone of ≈30 nm thickness. According to the simulations, the sputter yield for this system is ≈ 12 atoms per ion, i.e. in total one tenth of a monolayer is removed by the ion bombardment.

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
