# Peer review of "The thermoelectric conversion efficiency problem: Insights from the electron gas thermodynamics close to a phase transition"

_SciPost Physics, doi:SciPost Phys. Core 8, 015 (2025)_

## Round 2 · Referee Report · Anonymous (Referee 1) · 2022-8-4

Report

The manuscript “Thermodynamics of the thermoelectric working fluid close to the superconducting phase transition” by I. Khomchenko et al. is a study of the thermoelectric properties in a metal at the verge of a superconducting phase transition. According to the presented calculations an electronic system in such a state could offer a large enhancement of the thermoelectric energy conversion efficiency. To support their theoretical results, the authors performed measurements of the thermoelectric power in the 100-nm thick Ba(Fe1-xCox)2As2 thin film.
A relatively small efficiency of the currently used thermoelectric power generators drives intensive research to improve their thermal energy conversion ability. Since the figure of merit includes both the Seebeck coefficient (S) and the thermal conductivity, the effort is usually made to increase the former or supress the latter. In the present manuscript the authors chose the first path, concluding that near the superconducting transition S should be enhanced. As it has been already shown, both the transverse and longitudinal thermoelectrical coeffcients are expected to increase in the presence of fluctuating Cooper pairs [I. Ussishkin et al., Phys. Rev. Lett. 89, 287001 (2002)]. This applies to both 2D and 3D case. While the effect was shown to be likely responsible for the enhancement of the Nernst effect in the hight-Tc superconductors, the influence on the Seebeck was not noticed neither in bulk samples [e.g. S.D. Obertelli et al., Phys. Rev. B 46, 14928(R) (1992)] nor in thin films [e.g. H.-C. Ri et al., Phys. Rev. B 50, 3312 (1994), M. Putti et al., Int. J. Mod. Phys. B 17, 415 (2003)] of cuprates. The effect is absent also in the iron-based superconductors [I. Pallecchi et al., Supercond. Sci. Technol. 29, 073002 (2016)]. The authors tried to provide some references to experimental evidence, but “broadened” peaks from Refs. [26, 27, 28, 39, 40] look more like counterexamples. I mean that the Seebeck coefficient should not decrease when approaching critical temperature, if its magnitude were related to fluctuating Cooper pairs. Moreover, judging by the temperature where the absolute value of S start to increase in the given examples, the phenomenon is unlikely related to superconducting fluctuations. Another thing is that the sign of the excessive Seebeck coefficient, which I believe should be positive for a 2D case, is oppositely to that observed in Ba(Fe1-xCox)2As2. On the other hand, there is a good evidence of the enhancement of the thermoelectric power factor in ultra-thin FeSe [S. Shimizu et al., Nature Communications 10, 825 (2019)], but it was not related to presence of fluctuating Cooper pairs. It is also worth to notice that the effect was observed for samples thinner than ~15 nm and above this value films behaved like a bulk sample. In the present paper the 100 nm thick Ba(Fe1-xCox)2As2 layer is studied, which suggest that it should be treated rather as a 3D sample (especially as the anisotropy in this material is not large compared to, for example, cuprates).
My main complaint regarding the experimental part is that the authors present results obtained for just one sample, which was measured pristine and after ion bombardment. No dependence on the magnetic field or sample thickness is shown. Additionally, in the temperature range, where the authors claim the presence of superconducting fluctuations, I do not see any para-conductivity, which raises more doubts as to the interpretation given.
In conclusion, I think that without further support it is difficult to believe that the observed enhancement of Seebeck coefficient in Ba(Fe1-xCox)2As2 is due to fluctuating Cooper pairs. In addition, a main idea behind the studies has already been presented in Ref. [15]. Therefore, I recommend rejection of the manuscript.
At the end I have a few comments that perhaps might be useful for future submissions:
- The structure of the manuscript should be better organized.
- Abstract and introduction are way too long.
- Readability of Fig. 5 would benefit from changing scale from linear to logarithmic.
  • validity: -
  • significance: -
  • originality: -
  • clarity: -
  • formatting: -
  • grammar: -

Author:  Henni Ouerdane  on 2023-07-25  [id 3837]

(in reply to Report 1 on 2022-08-04)

First part of the comments by the Referee -- overall assessment and basic criticism

** Our reply:**

We thank the Reviewer for a very critical assessment of our work, which is useful , as we have sought to find a plausible interpretation of the experimental signatures we have found, especially the observed rise of the magnitude of the Seebeck coefficient starting near 50 K. In particular, we involved in these discussions a recognized expert on this material, Prof. Dr. Anna E. Boehmer, who is now also co-author of this paper. After intensive discussions, our interpretation is that the experimental signature in the Seebeck coefficient away from $T_{\rm c}$ could well be due to nematic fluctuations in the electron subsystem.

As a matter of fact, there is no universal model to simulate the thermoelectric properties of pnictide thin films across a wide range of temperatures. The approaches are essentially phenomenological. In our work we developed a thermodynamic model to discuss a particular experimental feature: the divergence of the power factor very close to $T_{\rm c}$, and in principle, the motif of fluctuations in the electron subsystem, at least in its phenomenology, translates very well from fluctuating Cooper pairs to nematic fluctuations. However, one has to say that for fluctuating Cooper pairs there have been established models for decades, whereas the mathematical description of nematic fluctuations is not yet at this established level. We have therefore chosen a compromise for the paper. In the theoretical description, we now make it clear that the crucial aspect comes from the fluctuation regime itself. In the experiment, we explain our data in terms of nematic fluctuations, which we substantiate with several references.

Next part of the comments by the Referee -- main complaint

** Our reply:**

We agree with the reviewer that consideration of fluctuating Cooper pairs alone here is insufficient. We have changed all text passages accordingly and interpret the data - in view of the literature - by nematic fluctuations.

In principle, we also agree with the reviewer that more experimental data would be necessary. However, this paper has a very strong theoretical ground. The experimental data should show that in principle the phenomenon of electronic fluctuations near phase transitions, shows up clearly in experimental transport data. A more detailed experimental study will follow in the future.

Third part of the comments by the Referee -- conclusion

** Our reply:**

The main idea behind Ref. [15] was to show quantitatively by means of numerical simulation of different models of electrically charged working fluids that if a system that can undergo a phase transition, its thermoelectric coupling strongly can be enhanced near the critical point. We illustrated this with the case of the 2D fluctuating Cooper pairs. The metric then introduced and used for comparison among different working fluids is the thermodynamic figure of merit $Z_{\rm th}T$, which is a combination of thermoelastic coefficients derived from the definition of the isentropic expansion factor $\gamma$.

In the present work, we significantly go beyond the scope of the previous work first by relating the thermodynamic figure of merit to the electronic thermoelectric figure of merit, i.e. the thermoelastic properties of the working fluid to its transport properties. To do so, we introduce an equivalent of the isentropic expansion factor based on transport coefficients, $\gamma_{\rm tr}$. The latter is a measure of the deviation from the thermostatic $\gamma$ when the system is out-of-equilibrium. In fact, thermoelectric conversion performance improves when $\gamma$ becomes large, which implies that the working fluid transport properties foster a high power factor if the working fluid's compressibility is larger. We see that $\gamma$ becomes large in the fluctuation regimes, and this is a nice illustration of the fluctuation-compressibility theorem in the context of thermoelectricity.

Further, contrary to Ref. [15], we present experimental data, although the development of a complete model that can addresses all aspects of the observed behaviors (Seebeck, electrical resistivity, power factor) in the clean and degraded samples is beyond the scope of the present work. What we had initially set out to do is to interpret and discuss the observed behavior close to $T_{\rm c}$ in light of our thermodynamic analysis, rather than provide a full model. It was clear that given the fact that we made a number of simplifying assumptions for clarity of the electronic working fluid models, we could not explain the trend starting at around 50 K down to close to $T_{\rm c}$. In the revised version, however, we provide a qualitative interpretation of the observed behavior in this range as a manifestation of nematic fluctuations.

We thus believe that in addition to the experimental data and their analysis, our thermodynamic study is not only non-trivial but also sheds light on the basic properties of the electron gas in thermoelectric materials. Further, we believe our work paves the way to the development of theoretical and experimental developments in the field of thermoelectricity as a number of questions remain to be answered.

Additional remarks by the Referee

** our reply:**

We have reshaped our manuscript to better report our work which has a large theoretical component.

True, the Introduction part is quite long, but we feel that it is necessary to properly lay out the rationale for this work, which is at the crossroads of various fields: thermodynamics, transport, solid-state physics, materials science, energy conversion. Same applies for the abstract that we tried to keep as concise yet as informative as possible.

Interestingly, the format of SciPost papers shows the content of the paper so any reader interested in particular aspects of the work can skip other parts that they find not essential for the understanding of the work. So, a long introduction provides more benefits than drawbacks here.

Close to $T_{\rm c}$ the rise is so sharp and the temperature range so small (to account for the limit of validity of the model) that a logarithmic scale does not provide a better view. In fact, this was our very first intent, but we decided that the linear scale provides a more satisfactory display of the curves.

---

## Round 2 · Referee Report · Anonymous (Referee 2) · 2022-9-2

Strengths

Interesting work. The experimental part is well explained.

Weaknesses

Theory part is badly explained. I have no reason to think there is anything wrong, but certain crucial information is missing (see report). This can be rectified by adding sentences at various points in the manuscript.

Report

Report on Khomchenko et al "Thermodynamics of the thermoelectric working fluid close to the superconducting phase transition"

This work has an experimental part and a theory part. I will discuss them separately.

EXPERIMENT: This is an interesting work which measure the thermoelectric properties of
superconducting thin-film made of Ba(Fe_{1−x} Co_x)_2 As_2 (with 100nm thickness). Under two conditions Fig 3a : higher structural quality (lower disorder) Fig 3b : lower structural quality (higher disorder)

The lower structural quality is created by the ion-bombardment of samples with higher structural quality; the difference between higher and lower structural quality is about 10%, That is to say that the ion-bombardment raises the resistivity of the thin-film's normal state at T ~ 30K (i.e. just above the superconducting transition) by about 10% from about 10 to 11 micro-Ohm-metres. Intriguingly, this modest change in structural quality has a huge effect on the thermoelectric response; the 10% reduction of the structural quality HALVES the thermoelectric response! That is to say that magnitude of the Seebeck coefficient at the superconducting transition changes from 200 to 100 microvolts/K.

Looking at the Seebeck curve in Fig 3b, it looks like a smoothed version of that in Fig 3a. Thus I speculate that the ion bombardment is causing the superconducting transition temperature, Tc, to vary randomly across the sample. It would be interesting to make a theoretical model in which each square of the sample has the alpha (T) of the shape in Fig 3a, but with Tc varying randomly from square to square. One could see if this reproduces the curve in Fig 3b.

THEORY: I found the theory part very hard to follow, the notation was confusing, and some crucial information seems to be missing. I list the things that confused me here, in the hope it will help the authors add sentences here and there to make the manuscript easier to understand.

(A) CONFUSION IN DEFINITION OF ZT VERSUS Z_{th}T

Eq. (1) tells us that ZT is the figure of merit including electrons AND phonons, and paragraph one of section 2.1 that Z_{th}T is that of the electrons alone (neglecting phonons). However in Eq. (2) Z_{th}T is defined via the isentropic expansion factor, and ZT becomes the figure of merit of the electrons alone in Eq. (3). This is very confusing, please fix the notation to be more consistent. I think one actually needs 3 different symbols for three different "ZT"s; - figure of merit including electrons AND phonons - figure of merit for electrons alone - figure of merit defined via the isentropic expansion factor

(B) MISSING INFORMATION FOR ZT

Unless I missed it, the model of the "2d electron gas" in Fig 4, 5 is incomplete. Firstly Fig 4 is misleading in implying that only kappa changes between the two plots, because alpha, sigma, C_{mu} and C_{N} are all also completely different for "2d electron gas" and "FCP". Then the appendices do not give the reader enough information to know kappa_{e}, alpha_{e}, etc, because the authors do not give the form of the terms inside Sigma(E) below eq. (16). The most common assumption for a 2d electron gas model for a thin-film of metal is that the temperature is much less than the Fermi energy, so Sigma is almost energy independent over the range of integration in Eq. (15) (Sommerfeld expansion), making the Seebeck coefficient extremely small. This does not seem to be the case here, but I cannot guess more without knowing what parameters are assumed. Specifically; (i) What is temperature range, Fermi energy, etc for "2d electron gas" plot in Fig 4? (ii) What is Fermi energy for "2DEG" plot in Fig 5, and "1D", "2D", "3D" plots in Fig. 6? Even better would be plots of alpha_{e}, kappa{e}, etc, for the authors' chosen parameters. (iii) What is the model for 0D in Fig 6,7? What is its density of states (DOS)?
Is it single-level system, with a delta-function DOS, or a Lorentzian-DOS due to broadening by coupling to leads? Or is it a multi-level system?

(C) MISSING INFORMATION FOR Z_{th}T FOR FLUCTUATING COOPER PAIRS (FCP).

Unless I missed it, the information enabling one to calculate Z_{th}T from Eq. (2) is incomplete for the fluctuating Cooper pairs (FCP).
At least, it is very hard to find the information in the manuscript. I see that appendix A contains formulas for some of the quantities that enter Eq. (2), but I did not find a formula for beta or C_{mu}. Similarly, I did not find a formula for N_{cp}, which enter Xi_{T} in Eq (12), necessary for Eq. (3). In contrast, I found the following formula for Z_{th}T for fluctuating Cooper pairs in Ref [15] (Ouerdane et al PRB, 2015) Z_{th}= ln[1/epsilon] Is that what is used here? If so, it would help the reader to give this formula!

(D) PHYSICS OF FLUCTUATING COOPER PAIRS

It would greatly help the reader if the authors included a paragraph that outlined the basic physics contained in the fluctuating cooper pairs (FCP) theory.

(E) CONDITION FOR NEGLECTING LATTICE PHONONS, kappa_{lat}

I am confused by whether it is reasonable to take kappa=kappa_{cp} rather than kappa=kappa_{cp}+kappa=kappa_{lat} in the right-hand plot in Fig 4. We see from Eq. (24) that kappa_{cp} goes to zero at the critical point, and so it will become less than kappa_{lat} at some point.
Have the authors estimated when this occurs, and made sure that it is reasonable to take kappa=kappa_{cp} for the temperature range in Fig 4? (As stated above, I could not find the range of temperature or epsilon for this plot in the manuscript).

Requested changes

See main report

  • validity: good
  • significance: good
  • originality: good
  • clarity: low
  • formatting: excellent
  • grammar: excellent

Author:  Henni Ouerdane  on 2023-07-25  [id 3836]

(in reply to Report 2 on 2022-09-02)
Category:
answer to question
reply to objection

We thank the Referee for an overall positive evaluation. We also take good note of the criticism and the various points raised, which we address in our reply.

** EXPERIMENT **

We appreciate a very in-depth look into and discussion of our experimental data. Since the manuscript has meanwhile become relatively long, in particular also due to necessary additions by the revision, we have not further deepened these very interesting aspects that can be considered in future works.

** THEORY**

(A) Confusion between $ZT$ and $Z_{\rm th}T$

** Our reply: **

The traditional notations used for the thermoelectric figure of merit are usually $ZT$ or $zT$, and they include the thermal conductivity of the lattice in the definition. In our work, as in some others where authors adapt their notations accordingly, we focus solely on the conversion performance of the electron gas. To avoid confusion we adopt the following notations in the revised version of the manuscript:

* $zT$ for the standard thermoelectric figure of merit, given by Eq. (1) of the manuscript;

* $z_{\rm e}T$ for the thermoelectric figure of merit \emph{disregarding} the phonon effects (i.e. without $\kappa_{\rm lat}$). It is a measure of the electron gas performance alone. Hence, the quantity $z_{\rm e}T$ includes only the Seebeck coefficient, the electrical conductivity, and the electronic thermal conductivity; and, of course, $z_{\rm e}T > zT$. In fact, in a given material, $z_{\rm e}T$ is an upper bound for $zT$ that would be reached in a situation where the so-called ``phonon glass'' (absence of heat transport by the lattice) would be managed. It is given by Eq. (2) of the revised manuscript;

* $Z_{\rm th}T$ is what we call the thermodynamic figure of merit defined from the electron gas heat capacity ratio, in Eq. (21). In the present work, we related the thermoelastic properties of the electrons to their transport properties -- see the parametric plots $z_{\rm e}T$ vs $Z_{\rm th}$ in Fig. 1 of the revised manuscript (Fig. 4 of the former version);

* $z_{\rm cp}T$ is the figure of merit of the 2D fluctuating Cooper pairs system, when the system's temperature $T$ is in the close vicinity of $T_{\rm c}$. It is given by Eq. (23);

* $Z_{\rm th,cp}T$ is what we call the thermodynamic figure of merit defined from the 2D FCP system's heat capacity ratio. It is given by Eq. (22).

(B) Missing information for $ZT$

** Our reply:**

In Fig. 4 of the former version of the manuscript (now Fig. 1), the $ZT$s'' and the$Z_{\rm th}T$s'' are those computed with the transport coefficients and thermoelastic coefficients of the 2D electron gas and the 2D FCP respectively. All the transport and thermoelastic coefficients used for the computation of the electron gas $ZT$'' (now $z_{\rm e}T$ and $Z_{\rm th}T$) are, indeed, completely different from those used to compute the 2D FCP$ZT$'' (now $z_{\rm cp}T$) and $Z_{\rm th}T$ (now $Z_{\rm th,cp}$). The same applies for the power factors in Fig. 5 of the former version of the manuscript (now Fig. 3), though we kept a generic notation as the arrows in the plot clearly indicate to what model the curves correspond.

The transport distribution function $\Sigma(E)$ has three terms: the relaxation time $\tau$, which for simplicity we take as constant, the velocity $v \propto \sqrt{E}$, and the density of states $g(E)$ for noninteracting electron systems. More detail is now given in Appendix A of the revised version.

For our numerical simulations, we consider low-density electron gases with concentrations $n_{\rm 3D}=10^{18} \mathrm{cm}^{-3}$, $n_{\rm 2D}=10^{12} \mathrm{cm}^{-2}$, and $n_{\rm 1D}=10^{6} \mathrm{cm}^{-1}$ for the three-, two-, and one-dimensional systems, respectively.

(i) The temperature ranges from close to $0$ K to $300$ K for the numerical calculations of the electron gases, and from $T_{\rm c}$ to $T_{\rm c}$ + 0.2 K for the 2D FCP.

(ii) The Fermi energy for the two-dimensional electron gas, is $E_{\rm F}^{\rm 2D}=2.39$ meV. The value of the Fermi energy in the three-dimensional case is $E_{\rm F}^{\rm 3D}=3.64$ meV, and $E_{\rm F}^{\rm 1D}=0.94$ meV for the one-dimensional case equals.

(iii) We use a single-level quantum dot to model 0D electron gas and a Lorentzian form for the density of states:

$g_{0D}(E) = \frac{\Gamma}{(E - E_{0})^{2} +(\Gamma/2)^{2}}$

where $\Gamma$ is the energy level width. The central energy of the channel is $E_{0}=0.99E_{F}^{\rm 2D}$, and the channel coupling energy $\Gamma = 0.1k_{B}T$.

(C) Missing information for $Z_{th}T$ for fluctuating Cooper pairs (FCP).

** Our reply:**

In the revised version, we transferred parts of the appendices of the previous version to the main text. In fact, as this work contains much material that pertains to thermodynamics, materials science, coupled transport, modeling, experiments etc., we wanted, for ease of reading, to avoid an overload of definitions and formulas that can be found in the cited published references.

To answer more specifically the point on the $Z_{\rm th}$ for the 2D FCP, we use and adapt the definition of $\gamma$ in Eq. (21) of the revised version, which explicitly reads:

$$ \gamma_{\rm cp} = 1 + \frac{\alpha_{\rm th,cp}}{\ell_{\rm cp}} = 1 +Z_{\rm th,cp}T $$

\noindent where $\ell_{\rm cp} = C_{N_{\rm cp}}/(q^2\chi_{T_{\rm cp}} T) = -C_{\rm GL} k_{\rm B}^2\ln\epsilon/q^2$, with $C_{\rm GL} = \hbar^2/(2m_{\rm cp}k_{\rm B}T_{\rm c}\xi^2)$ a dimensionless parameter in the Ginzburg-Landau free energy functional. Here $\xi$ is the coherence length. This is how we get to $Z_{\rm th,cp}T = \gamma_{\rm cp} - 1 = -\ln\epsilon$. See also the calculations shown in the appendix of Ref. [15] of the manuscript (Ouerdane et al PRB, 2015).

(D) Physics of fluctuating Cooper pairs

** Our reply:**

In the new section 2.2.3 of the revised manuscript, we give more information on the nature of the fluctuating Cooper pairs. The physics of fluctuating Cooper pairs is introduced and discussed at length in Ref. [42] of the revised manuscript: A. Larkin and A. Varlamov, Theory of Fluctuations in Superconductors, revised edition, (Oxford Science publications, 2009).

(E) Condition for neglecting lattice phonons $kappa_{lat}$

** Our reply:**

Interesting point here, which we now discuss in the new Section 2.4. Though lattice phonons are to be considered for a complete evaluation of a thermoelectric material's performance, they are primarily a cause of heat leaks, hence energy loss. It has been stated many times in thermoelectricity papers that the ideal thermoelectric material would behave as an electron crystal–phonon glass''\footnote{This paradigm was first expressed in G. A. Slack, CRC Handbook of Thermoelectrics, edited by D. M. Rowe (CRC Press, Boca Raton, FL, 1995), p. 407.} system, meaning high electrical conductivity and very low, nay zero thermal conductivity. So, more specifically, in the present work where we relate the thermoelastic properties to the transport properties of the electronic working fluid alone in Fig. 4 of the former manuscript (now Fig. 1 in the revised version with the new notations), we evaluate what the maximum conversion efficiency can ideally be in absence of heat leaks. In the case of 2D FCP, yes $\kappa_{\rm cp} \rightarrow 0$ in the limit $T \rightarrow T_{\rm c}$, but this does not pose any problem, quite the contrary in fact, as this shows what happens if the system would tend to become like anelectron crystal'', making $z_{\rm cp}T$ larger. Sure, accounting for $\kappa_{\rm lat}$ would negatively impact on the increase of $z_{\rm cp}T$, but here we see that without heat leaks, the conversion efficiency can tend to Carnot's efficiency in the limit $T\rightarrow T_{\rm c}$. The range over which the numerical calculations are performed in the normal phase is $\left]T_{\rm c} = 25.6 ~;~ 300 \right]$ K, and in the superconducting fluctuation regime $\left]T_{\rm c}~;~ T_{\rm c}+\delta T\right]$, with $\delta T = 0.2$ K.

---

## Round 3 · Referee Report · Alexei Vagov (Referee 3) · 2024-5-22

Strengths

1. The manuscript contains a fairly large amount of theoretical work and some experimental data also.
2. A basic thermodynamic analysis that gives very good insights into the thermoelectric conversion process as the thermoelastic properties and the transport properties are related. Parametric plots are shown.
3. A theoretical demonstration that in a 2D system near Tc the Wiedeman-Franz law is violated and that the power factor shows a diverging behavior as T goes closer to Tc.
4. Very interesting experimental results showing how the Seebeck coefficient and the electrical resistivity vary with temperature and that their combination into the power factor also diverges near Tc, thus showing a violation of the Wiedeman-Franz law as predicted by the model.
5. The experimental data is shown for a sample with high structural quality and the same sample with degraded structure because of ion bombardment.
6. An interesting discussion about the efficiency in "ideal cases" for simplified electron gas models. Only for the fluctuating regime near Tc, the efficiency can rise to Carnot efficiency.
7. A link between the compressibility of the electron gas and how it is efficient when it increases, which is the case when the gas shows density fluctuations.

Weaknesses

1. The lack of a model for thermoelectric conversion in the nematic fluctuation regime is a disadvantage for the manuscript, but given the complexity of the physical phenomena, such a model could be the object of a full separate work.
2. The manuscript is quite long and has several useful appendices. This shows that the authors want to give the reader as much detail as possible. That can make the reader lose sight of the main results, but given the complexity of the problem, a long text is unavoidable.

Report

The main idea of the manuscript is that the fluctuation regime close to a phase transition is beneficial for thermoelectric conversion efficiency. The thermodynamic analysis aims to explain why the transport coefficients near a phase transition allow for better thermoelectric transport. Some of the Authors already published the idea thermoelectric conversion near a superconducting phase transition in Ref. [15]. In their previous work, they focused on the thermoelastic properties of the conduction electron gas and showed that a quantity they call the thermodynamic figure of merit, which is more or less the isentropic expansion factor, diverges near the transition point. As for classical heat engines, using a working fluid that has a high heat capacity ratio, is beneficial for the heat-to-work conversion. In the mansucript, the authors relate the thermoelastic properties to the transport properties, with a focus on the power factor and the electronic zT (figure of merit without the contribution of the phonons to the thermal conductivity).
The manuscript contains a fairly large amount of theoretical work and some experimental data also. The authors explained in their replies to previous reviewers that the work is not a theory vs experiment, but a theoretical work to which experimental data has been added not to support the calculations but to support the idea that fluctuating regimes can enhance the conversion efficiency. The mathematical model is developped for two-dimensional fluctuating Cooper pairs very close to the superconducting transition temperature Tc and cannot be applied to interpret the experimental data that cover a wide temperature interval, and which shows an interesting behavior largely above Tc. The experimental data results from the measurement of the Seebeck coefficient and the electrical conductivity in a pnictide thin film with 100 nm thickness, which while not bulk is not 2D either. The authors suggest that the increase of the power factor is due to nematic fluctuations for which they provide no model to compute the thermoelastic properties and the transport properties. The suggest nonetheless that very close to Tc superconductive fluctuations may play a role.
The lack of a model for thermoelectric conversion in the nematic fluctuation regime is a disadvantage for the manuscript, but given the complexity of the physical phenomena, such a model could be the object of a full separate work.
What are we left with after reading the manuscript on the positive side:
- a basic thermodynamic analysis that gives very good insights into the thermoelectric conversion process as the thermoelastic properties and the transport properties are related. Parametric plots are shown.
- a theoretical demonstration that in a 2D system near Tc the Wiedeman-Franz law is violated and that the power factor shows a diverging behavior as T goes closer to Tc.
- very interesting experimental results showing how the Seebeck coefficient and the electrical resistivity vary with temperature and that their combination into the power factor also diverges near Tc, thus showing a violation of the Wiedeman-Franz law as predicted by the model.
- the experimental data is shown for a sample with high structural quality and the same sample with degraded structure because of ion bombardment.

- an interesting discussion about the efficiency in "ideal cases" for simplified electron gas models. Only for the fluctuating regime near Tc, the efficiency can rise to Carnot efficiency.
- a link between the compressibility of the electron gas and how it is efficient when it increases, which is the case when the gas shows density fluctuations.

On the negative side: the lack of a model to better support the description and interpretation of the experimental data.
The manuscript is quite long and has several useful appendices. This shows that the authors want to give the reader as much detail as possible. That can make the reader lose sight of the main results, but given the complexity of the problem, a long text is unavoidable.
I believe that thermoelectricity with phase transitions in the conduction electron gas can provide new valuable theoretical problems to consider. The nematic phase transition and the fluctuating regime can be the object of interesting works. From the experimental viewpoint, this work can also inspire new work where for example very thin films or 2D materials are studied.
The authors might comment on this recent paper: Nat Commun 15, 776 (2024) doi: 10.1038/s41467-024-45093-6 by Zhao et al. where the authors work on the modeling of critical thermoelectric transports.
I recommend publication of the manuscript as the physics is interesting and well discussed, and because it provides good ground for future theoretical works and perhaps experimental work, which will fill the gaps of this manuscript.

Requested changes

The authors might comment on this recent paper: Nat Commun 15, 776 (2024) doi: 10.1038/s41467-024-45093-6 by Zhao et al. where the authors work on the modeling of critical thermoelectric transports.

Recommendation

Publish (easily meets expectations and criteria for this Journal; among top 50%)

  • validity: top
  • significance: high
  • originality: top
  • clarity: top
  • formatting: excellent
  • grammar: excellent

Author:  Henni Ouerdane  on 2024-10-15  [id 4868]

(in reply to Report 1 by Alexei Vagov on 2024-05-22)

Dear Professor Vagov,

On behalf of my coauthors, I thank you for your supportive comments and recommendation for publication. Here, I provide a reply to the points your raised.

Concerning weaknesses:

  1. The lack of a model for thermoelectric conversion in the nematic fluctuation regime is a disadvantage for the manuscript, but given the complexity of the physical phenomena, such a model could be the object of a full separate work.

2. The manuscript is quite long and has several useful appendices. This shows that the authors want to give the reader as much detail as possible. That can make the reader lose sight of the main results, but given the complexity of the problem, a long text is unavoidable.

Our reply:

1.We see that more as a limitation of the scope of our work than a disadvantage though we understand that having such a model would strengthen the manuscript. In fact, while we could perform calculations using the works of Varlamov and Larkin [Theory of Fluctuations in Superconductors, Oxford University Press (2005)] for 2D fluctuating Cooper pairs systems, establishing a transport theory in the nematic fluctuation regime is indeed a task beyond the scope of the present work, which initially was meant to be restricted to a thermodynamic regime very close to $T_{\rm c}$ and away in the normal regime where models of the noninteracting electron gas are applicable. We agree that there is a need to have a dedicated model able to describe the experimental data we have above $T_{\rm c}$, beyond the superconducting fluctuating regime and where noninteracting electron gas models are no longer adequate, like in the nematic fluctuation regime. Instead, we described the temperature-dependent Seebeck coefficient and electrical conductivity data, and discussed their behavior assuming that nematic fluctuations play an important role. While this is insufficient for a fully-fledged theory vs experiment type of paper, our manuscript is not quite of such a type and already contains much physics, simulation and experimental results, which allow to explain that fluctuating regimes close to a phase transition can foster a sizeable increase of the thermoelectric energy conversion efficiency.

2.The first version v1 (available on arXiv) was much shorter and as we wrote the version v2, we saw a need to extend the paper to provide more information in the main text for clarity, referring to our previous work [Phys. Rev. B 91, 100501(R) (2015)]. It turned out that one of the reviewers of the version v2 did not find the thermodynamic approach clear enough and considered this as a weakness [Report 2 on 2022-9-2]:

Theory part is badly explained. I have no reason to think there is anything wrong, but certain crucial information is missing [...],

so we extended the presentation of the theory part. As we rewrote the manuscript, we attempted to produce a text that would not be misleading in the sense that the paper is not an actual theory vs experiment one, and we meant to highlight the limit of validity for the use of the 2D fluctuating Cooper pairs model. To ease the reading, a detailed table of content is available just below the abstract so that any reader at a glance can get a overall grasp of the work we report in the manuscript.

Requested changes

The authors might comment on this recent paper: Nat Commun 15, 776 (2024) doi: 10.1038/s41467-024-45093-6 by Zhao et al. where the authors work on the modeling of critical thermoelectric transports.

Our reply

Thank you for mentioning this very recent paper that also considers that critical phenomena can be beneficial for thermoelectric energy conversion. The authors claim that a quantitative and comprehensive model of thermoelectricity close to a phase transition is still lacking, and rightly so. Their work is a contribution to fill that gap with a strong focus on the provision of formulas to compute transport coefficients using the Landau theory and the Boltzmann transport equation to make their model tractable. They get quantitative results but not a comprehensive model. In their work, the authors consider structural phase transitions and the effects of band broadening and carrier-soft TO phonon interactions on the Seebeck coefficient and on the electrical conductivity respectively. The lambda shape they obtain for the Seebeck and the electrical conductivity, with a peak at the critical point confirms that phase transitions foster the desired transport properties for thermoelectric energy conversion efficiency. Interestingly, the band broadening is caused by structural fluctuations near the critical phase transition temperature.

In our work, we tackle the problem of thermoelectricity close to a phase transition of the conduction electron gas (working fluid) with a thermodynamic analysis. Unlike in the work reported in Nat Commun 15, 776 (2024), we do not consider structural phase transitions but the fluctuation regimes in the electron gas above the superconducting and the nematic phase transition temperature. We seek to find what regime can enhance the electron gas isentropic expansion factor first, and we try and see if that correlates with an improvement of the transport coefficients. While in the paper mentioned above they calculate zT accounting for the thermal conductivity of the lattice, in our work we consider the power factor. Our theoretical model shows that in the fluctuating regime in the close vicinity of the superconducting phase transition the power factor can be greatly enhanced. Our experimental results, for which there is a need for a dedicated model, show that in the nematic fluctuating regime the power factor is enhanced too. The isentropic expansion factor increase is due to the increase of the compressibility of the electron gas in the fluctuating regime.

To summarize, while the types of critical phenomena considered in our manuscript and in Nat Commun 15, 776 (2024), are different, the rationale is the same: placing a thermoelectric material in a thermodynamic regime close to a phase transition can favor a strong enhancement of its energy conversion performance due to the influence of critical phenomena on the transport coefficients. Note that it would be interesting to compute the temperature dependence of the isentropic expansion factors of the electron gas in Cu2Se and in Cu2Se{1-x}Sx, accounting for band broadening and carrier-soft TO phonon interactions and see theoretically how each of these two phenomena influence it when combined and when separated.

We now cite this paper in the Introduction section.

Author:  Henni Ouerdane  on 2025-01-31  [id 5172]

(in reply to Henni Ouerdane on 2024-10-15 [id 4868])
Category:
remark

Dear Referee,

Please note that for unclear reasons, the reply that I posted on October 15, 2024, were vetted only on January 30, 2025. I have no idea why it took 3,5 months for the vetting to be done, even more so that I sent reminders.

Further, the system did not allow me to submit the revised version that contains all the changes according to your comments and criticism. The revised version is v4 on the arXiv website, which was posted on October 14, 2024.

If you would like to read the revised version, please consider the latest arXiv version (October 14, 2024) rather than the previous one you already reviewed accessible on the SciPost website.

Thank you again for your time and useful reviewing work. I am glad that at long last, you may read my reply.

Sincerely,

Henni

---

## Round 3 · Referee Report · Anonymous (Referee 1) · 2024-6-11

Strengths

This work merits publication., because I believe that the experiment is new, even if I recently discovered that nearly all the theory is already published in
"Enhanced thermoelectric coupling near electronic phase transition: The role of fluctuation Cooper pairs", Henni Ouerdane, Andrey A. Varlamov, Alexey V. Kavokin, Christophe Goupil, and Cronin B. Vining, Phys. Rev. B 91, 100501(R) (2015).

Weaknesses

The manuscript is not clear about whether there is good agreement between the theory and the experiment, or not. Or indeed, maybe there are regimes where they agree and regimes where they do not; the manuscript does not say.

It is odd to present the theory (already published elsewhere) in so much detail unless one wants to make careful quantitative fits between theory and experiment, extracting and explaining the relevant fitting parameters.

This weakness is immediately overcome if the authors add a fit of the theory to the experimental data for alpha in Fig 2, and include an explanation of how the fitting was done, what fitting parameters that reveals, and what those parameters imply (see point A of my report below).

Report

Before starting this report, I want to say that I fear that I am not reviewing the latest version of this manuscript ! This is because Henni Ouerdane's response to previous report 2 (at link: https://scipost.org/submissions/2110.11000v2/#comment_id3836) listed a bunch of changes in the manuscript, but those changes are NOT in the version that I have access to via the SciPost website. To be specific, the SciPost website directs me to the version here (https://arxiv.org/abs/2110.11000v3), but it that does not contain info that the response to previous report 2 said is there,
such as:
- Fermi energies such as E_F^{1D}= 0.94meV, E_F^{3D}=3.64meV, etc.
- Density of states for 0D.
- the energy dependence of the velocity.
Thus I suspect that this is not the most recent version of the manuscript. That means that some of my comments below may be out-of-date. However I give them all here, and suggest that the authors ignore any comments that are not relevant to the most recent version of the manuscript.

MAIN COMMENT:

I cannot recommend this work for publication until the authors address the following two comments.

(A) The experimental work is of value, but nearly all of the theoretical part (for normal state and fluctuating cooper pairs) seems to be reproduced from an earlier work by the same authors:
"Enhanced thermoelectric coupling near electronic phase transition: The role of fluctuation Cooper pairs,", Henni Ouerdane, Andrey A. Varlamov, Alexey V. Kavokin, Christophe Goupil, and Cronin B. Vining, Phys. Rev. B 91, 100501(R) (2015).
This reproducing of earlier theory would be worthwhile if the goal is to fit that theory to the experiment. However, the manuscript gives no indication how the theory fits the experiment. This must be rectified.

Fig 2 gives an experimental curve for thermoelectric response, alpha, but it does not appear to look like the theory given above Eq. (23), which predicts alpha_{cp} ~ ln[epsilon] = ln[ln[T/Tc]].
Is the reader supposed to understand that it is a more complicated function of T? If the experiment should be fitted by a sum of alpha_{cp} and alpha_{e}),
then the manuscript need to explain this.

In short, do the theories presented in the earlier sections describe the experiment in Fig 2 or not Provide the fit, and the fitting parameters, please!

Fig 3 shows theory curves on top of the experimental plot of alpha^2 sigma, where sigma is conductivity. The divergence of alpha^2 sigma is fitted by the theory, however I believe that this divergence is entirely due to the divergence of conductivity at T=Tc,
and nothing to do with the thermoelectric response, alpha. Thus this fit is a poor test of a theory trying to explain the thermoelectric response, it would be much more discriminating test to fit alpha in Fig 2.

(B) I find the discussion in section 4.2 is confusing, because it never mentions the contradiction between using Z_{th}T or Z_{e}T to define the thermodynamic efficiency of a thermoelectric machine.
It states that the thermodynamic efficiency is given by Eq. (25) which depends on Z_{th}T (recalling that gamma= 1+Z_{th}T). But the manuscript does not mention that this contradicts the standard approaches, which state that the thermodynamic efficiency is given by the same formula with Z_{e}T or Z_{cp}T INSTEAD of Z_{th}T or Z_{th,cp}T (see e.g. the review of Benenti et al (2017) = Ref [75] of this manuscript).

The earlier parts of the manuscript (specifically Fig 1) shows that Z_{th}T is different from ZT, and this difference is up to five orders of magnitude (for fluctuating Cooper pairs).
Hence, the formula in Eq. (25) will give completely different results from the standard formula.
So which is correct?
I would like the manuscript to discuss the contradiction,
and give arguments why one may be better than the other.

My personal opinion is that Z_{th}T and Z_{th,cp}T are WRONG for predicting the thermodynamic efficiency of a real thermoelectric system, and Z_{e}T and Z_{cp}T are CORRECT (assuming phonon effects are neglected).
However, I am not sure about this, so I would appreciate any arguments that the authors have about which one is correct.

MINOR COMMENTS ON THE PRESENTATION

Addressing the following comments will greatly help the reader. However, I leave it up to the authors to decide how to do so.

(1) There is no reference to appendix A in the main text. I recommend that the authors should add two equations to section 2.2.2, with Z_{th}T and Z_{e}T, stating that Z_{e}T is calculated in appendix A.
Placing these two equations, Z_{th}T and Z_{e}T, one after the other would allow readers to see the similarities and differences, and so allow them to appreciate Fig 1a (a figure which currently has no explanation).

(2) The manuscript does not give all parameters necessary to understand Figs 1 and 5. One essential difference between Z_{th}T and Z_{e}T is that

- Z_{th}T depends on the E dependence of the density of states g(E)
- Z_{e}T depends on the E dependence of tau(E) (v(E))^2 g(E)

Hence the shape of the plots in Figs 1 and 5 depend critically on the choice of energy dependence of tau(E) (v(E))^2. Different choices of tau(E) (v(E))^2 will give very different plots. Appendix A assumes that tau(E) is independent of E, which seems reasonable. However, the energy-dependence of v(E) is intimately related to g(E), but it also depends on the choice of band-structure (e.g. parabolic bands or something else). Hence it would really help the reader to explicitly explain this, state what assumptions are made (parabolic bands, or something else).

(3) If I assume parabolic bands, then
v(E) ~ E^{1/2} (as in H. Ouerdane's response to previous report 2)
d(E) ~ E^{(d/2-1}
Hence assuming tau(E) is E-independent as stated in appendix A, the term in the integrands for Z_{e}T is
tau(E) (v(E))^2 g(E) ~ E^{d/2},
while the equivalent term in integrands for Z_{th}T is
g(E) ~ E^{d/2-1}.
Thus, the integrands in the two quantities always differ by a single power of E. Hence, it seems paradoxical that Fig 5 shows a straight line indicating Z_{e}T=Z_{th}T in 1D, when the integrals are clearly different. The authors should explain the details of how that happens.

(3) The authors must add the explanation of 0D to the manuscript, the fact it is a quantum dot with a Lorentizian transmission, as explained in the response to the previous report 2.

(4)It is also worth adding a comment to the manuscript that mentions the other difference between Z_{th}T and Z_{e}T. The denominator of Z_{e}T contains an extract term proportional the square of the thermoelectric response (the L_{21}L_{12} term in equation for kappa_{e} in Eq. (31)). There is no analogue of this term in Z_{th}T.

(5) I cannot find the place in the manuscript that gives the value of the electro-chemical potential for the examples in Fig. 4 (these values were given in the response to previous report 2). These values are crucial to understand the curves in Fig 4,
because for given d(E) and v(E), the parameter that matters in the ratio of temperature to electro-chemical potential. For example, if one takes a sample with more charge carriers per unit volume so its electro-chemical potential is larger (e.g. larger Fermi energy), then one needs a larger temperature to achieve the same Z_{e}T (and hence achieve the same maximum thermoelectric conversion efficiency).

Requested changes

See my report.

Points A and B are strong recommendations, I cannot recommend publication before they are addressed.

The other points are optional, but they should be easy to do, and would be a great help to readers.

Recommendation

Ask for minor revision

  • validity: high
  • significance: ok
  • originality: ok
  • clarity: ok
  • formatting: good
  • grammar: good

Author:  Henni Ouerdane  on 2024-10-15  [id 4869]

(in reply to Report 2 on 2024-06-11)

Dear Referee,

Thank you very much for your report, comments and criticism. I provide our reply below.

Weaknesses

1.The manuscript is not clear about whether there is good agreement between the theory and the experiment, or not. Or indeed, maybe there are regimes where they agree and regimes where they do not; the manuscript does not say.

2.It is odd to present the theory (already published elsewhere) in so much detail unless one wants to make careful quantitative fits between theory and experiment, extracting and explaining the relevant fitting parameters.

3.This weakness is immediately overcome if the authors add a fit of the theory to the experimental data for alpha in Fig 2, and include an explanation of how the fitting was done, what fitting parameters that reveals, and what those parameters imply (see point A of my report below).

Reply:

1.The main hypothesis of our work is that fluctuating regimes near a phase transition foster a sizeable increase of the thermoelectric conversion efficiency. We show this considering on the one hand a theoretical model of a 2D fluctuating Copper pairs gas close to $T_{\rm c}$, and on the other hand with experimental data that we aim to fit only close to $T_{\rm c}$ where the model is supposed to be applicable. As explained in the main text, we are unable to account for the observed trends for the Seebeck and power factor starting from 50 K down to $\approx T_{\rm c}$. Our interpretation is that nematic fluctuations can play a role here.

Hence, since its initial version v1, available on arXiv, the model, based on a simple tractable approach, is meant to show, considering the superconducting phase transition, that critical phenomena may be of interest to increase the thermoelectric conversion efficiency; and it is used to partly interpret the observed behavior of the power factor obtained from the experimental data \emph{only} in a very restricted temperature range $[T_{\rm c}; T_{\rm c} + \delta T]$, with $\delta T \ll 1$ K, where the fluctuation regime extremely close to the superconducting phase transition.

The 2D fluctuating Cooper pair model cannot be used to explain the power factor growth with the temperature decrease, from its onset at around 50 K down to temperatures close to $T_{\rm c}$ but outside the superconducting fluctuating regime, and we do not claim it can. That is why in v3, we discuss the possibility of the effects of the nematic fluctuations to describe the observed behavior, though we currently do not have a model of transport coefficients in the nematic fluctuating regime.

2.The version v1 was much shorter than the subsequent ones as the theory part was meant to include a a brief recap of previous works (which we systematically cited) together with comments aiming at clarity, as well as new theory content on the correlation between the thermoelastic properties of the electron gas and its transport properties, and how this relates to thermoelectric conversion performance evaluation. Some details were either in appendixes or in the cited papers, notably in [Phys. Rev. B 91, 100501(R) (2015)]. We decided that for ease of reading of the theoretical part, the main text in the manuscript should be completed with more theoretical elements in the version v2. However, it turned out, as explained in our reply to Prof. Vagov's comments [Report on 2024-5-31], that one of the reviewers of the version v2 did not find the thermodynamic approach clear enough, and considered this as a weakness [Report 2 on 2022-9-2]:

"Theory part is badly explained. I have no reason to think there is anything wrong, but certain crucial information is missing [...]",

so we reshaped and extended the presentation of the theory part in the version v3, hence your comment:

"I recently discovered that nearly all the theory is already published."

That said, it is important to notice that in [Phys. Rev. B 91, 100501(R) (2015)], the focus is only on the thermoelastic properties of noninteracting electron systems and those of the 2D fluctuating Cooper pairs. In the current work, which extends the scope of the previous one, we aim to understand what thermodynamic conditions may favor the desired transport properties and we establish a correlation between the thermoelastic properties and the transport properties: the larger $Z_{\rm th}T$, the larger $zT$, which confirms the hypothesis that a large electronic isentropic expansion factor favors the energy conversion efficiency in thermoelectric systems. We also introduce an analogue of the isentropic expansion factor $\gamma_{\rm tr}$ for an out-of-equilibrium system. The new theoretical material is definitely not marginal.

As shown in Fig. 3 of the manuscript (version v3), the power factor obtained from the measurement data undergoes a very steep increase as the temperature $T$ decreases and approaches $T_{\rm c}$. The 2D fluctuating Cooper pair model, which is only valid for $T$ close to $T_{\rm c}$ yields a power factor $\sigma_{\rm cp}\alpha_{\rm cp}^2$ proportional to $\ln^2(\varepsilon)/\varepsilon$, with $\varepsilon = (T - T_{\rm c})/T_{\rm c}$. Again, in the limit of validity of the model, i.e. as $\varepsilon \rightarrow 0$, the calculated power factor diverges. On Fig. 3 of v3, we indicate with an arrow the quasi vertical line, which has been computed with the formula of the power factor $\sigma_{\rm cp}\alpha_{\rm cp}^2$ scaled with the value of the power factor at 300 K (which is obtained using the 2D electron gas model at room temperature)..

3.The only fit of the Seebeck coefficient that we can do to a good approximation is for $T\approx T_{\rm c}$ using the following formula

$$\alpha_{\rm cp} = (\alpha_{\rm GL}k_{\rm B}/2e)\ln\varepsilon$$
where $\varepsilon = \ln(T/T_{\rm c})$ and $\alpha_{\rm GL}$ is a constant.

And for temperatures $T$ in the non-degenerate regime, we can approximately fit the Seebeck coefficient with the formula

$$\alpha = L_{12}/(qTL_{11})$$
with $L_{ij}$ being Onsager's kinetic coefficients calculated with parameters given in the manuscript appendix. We cannot do a fit in between these two regimes at present. However, note that our main focus is on the observed behavior near $T_{\rm c}$.

Main Comments

Preamble

the version [v3] does not contain info that the response to previous report 2 said is there, such as: - Fermi energies such as $E_F^{1D}$= 0.94 meV, $E_F^{3D}$=3.64 meV, etc. - Density of states for 0D. - the energy dependence of the velocity.

Reply: Formulas and data are now provided in full in the Appendix A.

Point (A):

1.The experimental work is of value, but nearly all of the theoretical part (for normal state and fluctuating cooper pairs) seems to be reproduced from an earlier work by the same authors: "Enhanced thermoelectric coupling near electronic phase transition: The role of fluctuation Cooper pairs,", Henni Ouerdane, Andrey A. Varlamov, Alexey V. Kavokin, Christophe Goupil, and Cronin B. Vining, Phys. Rev. B 91, 100501(R) (2015). This reproducing of earlier theory would be worthwhile if the goal is to fit that theory to the experiment. However, the manuscript gives no indication how the theory fits the experiment. This must be rectified.

2.Fig 2 gives an experimental curve for thermoelectric response, alpha, but it does not appear to look like the theory given above Eq. (23), which predicts $alpha_{cp} ~ \ln[epsilon] = \ln[\ln[T/Tc]]$. Is the reader supposed to understand that it is a more complicated function of T? If the experiment should be fitted by a sum of $alpha_{cp}$ and $alpha_{e}$), then the manuscript need to explain this. In short, do the theories presented in the earlier sections describe the experiment in Fig 2 or not Provide the fit, and the fitting parameters, please!

3. Fig 3 shows theory curves on top of the experimental plot of $\alpha^2 \sigma$, where $\sigma$ is conductivity. The divergence of $\alpha^2 \sigma$ is fitted by the theory, however I believe that this divergence is entirely due to the divergence of conductivity at $T=T_c$, and nothing to do with the thermoelectric response, $\alpha$. Thus this fit is a poor test of a theory trying to explain the thermoelectric response, it would be much more discriminating test to fit $\alpha$ in Fig 2.

Reply:

1.We explained above how and why the writing of the theoretical section evolved since v1. Note that all notions and results already introduced and discussed in [Phys. Rev. B 91, 100501(R) (2015)] have been systematically referred to the 2015 paper, in the manuscript versions v1, v2, and v3. The theoretical model has also been more thoroughly presented and, importantly, extended in relation to the transport properties and conversion efficiency calculations, which were missing in the 2015 paper.

The main goal of the theory part is two-fold: 1/ to show with a thermodynamics approach that no material and no system for which noninteracting electron gas models provide a good description, can be good candidates for high-efficiency thermoelectric conversion even under ``ideal'' working conditions (no detrimental phonon effects and no other source of dissipation); 2/ to show that harnessing thermoelectric energy conversion near critical phenomena, like, e.g., the superconducting fluctuating regime in 2D systems, could be a promising venue in terms of efficiency. Calculations show that for the latter, efficiency can be high, i.e. approach the Carnot efficiency, albeit in a very restricted range of temperature.

The analysis of the experimental data across the full temperature range, from the critical temperature $T_{\rm c}$ to 300 K, necessitates more sophisticated models to account notably for the band structure, the electronic density of states, interaction with phonons, and the thickness of the thin film (which does not make it a 2D system, but not bulk either). In the manuscript, we suggest an interpretation of the increase of the power factor, which stems from the increase of the thermoelectric coupling: below 50 K nematic fluctuations can play a role, and close to the superconducting phase transition fluctuating Cooper pairs also play a role, albeit in a very small temperature range above $T_{\rm c}$.

As explained in previous replies, our work is based on the hypothesis that fluctuating regimes can be beneficial for thermoelectric conversion efficiency. We showed this, focusing on the electron gas alone with a tractable theoretical model of 2D fluctuating Cooper pairs valid in a very restricted temperature range, and, independently, with experimental results that also indicate that nematic fluctuations could play a role well above $T_{\rm c}$. As mentioned in the Conclusion section, we relate our calculations and interpretations to the fluctuation-compressibility theorem: the larger the fluctuations are, the larger the compressibility of the working fluid is, and, in turn, the larger the isentropic expansion factor. The thermodynamic analysis adapted to the thermoelectric problem shows that to increase the figure of merit $zT$, one has to find the conditions for the electron gas to be a good working fluid.

2.We may provide a fit of the Seebeck coefficient $\alpha$ only in a very narrow range above $T_{\rm c}$, where we assume that fluctuating Cooper pairs play a dominant role. Given the scale of Fig. 2, the simulated curve would appear as a very small vertical segment near $T_{\rm c}$. A zoom on that part would simply reveal a logarithmic behavior. The formula is given in the main text, section 2.4: $\alpha_{\rm cp} = \nabla \mu/q\nabla T = \alpha_{\rm GL} k_{\rm B} \ln\varepsilon/2e$, valid in the $[T_{\rm c}; T_{\rm c} + \delta T]$, with $\delta T \ll 1$ K and the parameters in the Appendix A. A realistic fit beyond $T_{\rm c} + \delta T$ necessitates to compute $\alpha$ using the system's electronic band structure and density of states. Clearly, the ferropnictide thin film experimentally studied in the manuscript cannot be described by a simple noninteracting electron gas model with parabolic bands. That said, one key simulation result is the quasi-vertical line over a tiny temperature range in the vicinity of $T_{\rm c}$ (our temperature regime of interest) in Figure 3 as we want to highlight the influence of critical phenomena on the power factor.

3.We disagree with you here. Figures 2 and 3 show experimental results across a large temperature range for $\alpha(T)$, $\rho(T)\equiv 1/\sigma(T)$, and $\sigma(T)\alpha^2(T)$, under two thin film sample structural conditions: high-structural quality, and low-structural quality after ion bombardment. The degraded sample shows in Fig. 2b that as $T\rightarrow T_{\rm c}$, $\rho(T) \rightarrow 0$, and hence that $\sigma(T)$ diverges while $\alpha(T)$ barely increases in magnitude before saturation. On Fig. 3, the degraded sample shows that the power factor $\sigma(T)\alpha^2(T)$ remains finite too in spite of a resistivity that goes to 0 as $T\rightarrow T_{\rm c}$. Conversely, Fig. 3 shows that the power factor for the high-structural-quality sample increases very largely when as $T\rightarrow T_{\rm c}$, which is, as shown on Fig. 2, thanks to the large increase of the Seebeck (which is modelled theoretically with a formula that shows a diverging as $T\rightarrow T_{\rm c}$). If the Seebeck had no effect on the observed behavior, one could suppose that the diverging resistivity alone would allow for a steep increase of the power factor, which is not the case here. Therefore the effect of the large increase of the Seebeck coefficient on the power factor, is shown by the measurement data by direct comparison of the high-structural quality and the low-structural quality cases. The theoretical fit close to $T_{\rm c}$ is relevant. The observed behavior has indeed something to do with the thermoelectric response in the close vicinity of $T_{\rm c}$.

Point (B)

1.I find the discussion in section 4.2 is confusing, because it never mentions the contradiction between using $Z_{th}T$ or $Z_{e}T$ to define the thermodynamic efficiency of a thermoelectric machine. It states that the thermodynamic efficiency is given by Eq. (25) which depends on $Z_{th}T$ (recalling that $\gamma= 1+Z_{th}T$). But the manuscript does not mention that this contradicts the standard approaches, which state that the thermodynamic efficiency is given by the same formula with $Z_{e}T$ or $Z_{cp}T$ INSTEAD of $Z_{th}T$ or $Z_{th,cp}T$ (see e.g. the review of Benenti et al (2017) = Ref [75] of this manuscript).

2.The earlier parts of the manuscript (specifically Fig 1) shows that $Z_{th}T$ is different from $ZT$, and this difference is up to five orders of magnitude (for fluctuating Cooper pairs). Hence, the formula in Eq. (25) will give completely different results from the standard formula. So which is correct? I would like the manuscript to discuss the contradiction, and give arguments why one may be better than the other.

3.My personal opinion is that $Z_{th}T$ and $Z_{th,cp}T$ are WRONG for predicting the thermodynamic efficiency of a real thermoelectric system, and $Z_{e}T$ and $Z_{cp}T$ are CORRECT (assuming phonon effects are neglected). However, I am not sure about this, so I would appreciate any arguments that the authors have about which one is correct.

Reply:

1.The quantities $Z_{\rm th}T$ and $Z_{\rm th,cp}T$ are not contradictory to $Z_{\rm e}T$ and $Z_{\rm cp}T$ respectively, but perfectly complementary. We developed a thermodynamic approach to gain more insights into the thermodynamic conditions that may favor a significant increase of the thermoelectric conversion efficiency. Using $Z_{\rm th}T$ and $Z_{\rm th,cp}T$ for a study of the electron gas, here considered as a working fluid, is definitely appropriate. Saying that this is wrong amounts to stating that the isentropic expansion factor is a meaningless quantity. In fact, the larger $Z_{\rm th}T$ and $Z_{\rm th,cp}T$ are, the less entropy is produced during the energy conversion, which is exactly what one wants. The analysis of $Z_{\rm th}T$ and $Z_{\rm th,cp}T$ points to the most favorable conditions to minimize entropy production. Once these are known, one may study transport and conversion under these conditions. The correlations between $Z_{\rm th}T$ and $z_{\rm e}T$ on the one hand, and $Z_{\rm th,cp}T$ and $z_{\rm cp}T$ on the other hand, depicted in Fig.~1, show that all these quantities are not contradictory.

2.While the maximum theoretical efficiency which can be reached can be calculated with the traditional formula, we show that from the thermodynamic viewpoint, knowledge of the thermoelastic coefficients of the working fluid, here the electron gas, yields the same ranges of efficiency. In fact, the formulas for the maximum efficiency using $Z_{\rm th}T$ and $Z_{\rm th,cp}T$ on the one hand, and $Z_{\rm e}T$ and $Z_{\rm cp}T$, are formally the same -- See Eqs. (25) and (26) of the revised manuscript. This is shown and discussed in the revised Section 4.2. The results are very similar as shown on the revised Fig. 4.

3.As discussed just above the use of $Z_{\rm th}T$ and $Z_{\rm th,cp}T$ is not wrong at all; it gives additional insights into the physics of thermoelectric energy conversion. Both approaches are correct. We hope that the above clarifies the matter.

Minor comments on the presentation

There is no reference to appendix A in the main text. I recommend that the authors should add two equations to section 2.2.2, with $Z_{th}T$ and $Z_{e}T$, stating that $Z_{e}T$ is calculated in appendix A. Placing these two equations, $Z_{th}T$ and $Z_{e}T$, one after the other would allow readers to see the similarities and differences, and so allow them to appreciate Fig 1a (a figure which currently has no explanation).

Reply: Actually, there was a reference to Appendix A in the main text, page 9, in the sentence just above the section Results. But following the comment, we found it better to rather have references to Appendix A, below Eq. (2), below Eq. (15) and in the caption of Fig.~1.

$Z_{\rm th}T$ for the systems of noninteracting electrons is already given by Eq. (21) in Section 2.3 devoted to the thermodynamic figure of merit, and the formulas of the thermoelastic coefficients to calculate it given in Eqs. (13), (14) and (15) in Section 2.2.2 where it is announced that they will be used for that purpose.

$z_{\rm e}T$ is also already given by Eq. (2) in the preamble of the section Theory. We note though that we could have referred to Appendix A to indicate that expressions for the transport coefficients are given there. This is now done. We have also modified Eq. (2) including a definition of $z_{\rm e}T$ with the Seebeck coefficient and the Lorenz number.

Just below Eq. (21) we mention the thermoelastic counterpart $\ell$ of the Lorenz number $L$ and discuss their physical interpretation as $\ell$ enters the definition of $Z_{\rm th}T$ and $L$, the definition of $z_{\rm e}T$ as shown in the main text: $Z_{\rm th}T = \alpha_{\rm th}^2/\ell$ and $z_{\rm e}T = \alpha^2/L$.

In the case of the 2D fluctuation Cooper pairs, $Z_{\rm th,cp}T$ and $z_{\rm cp}T$ are given by Eqs. (22) and (23) and one sees that the main difference is in the dependence on $\varepsilon$, the latter having a prefactor $1/\varepsilon$. We also comment on it now in the text below Eq. (23).

The manuscript does not give all parameters necessary to understand Figs 1 and 5. One essential difference between $Z_{th}T$ and $Z_{e}T$ is that - $Z_{th}T$ depends on the E dependence of the density of states $g(E)$ - $Z_{e}T$ depends on the $E$ dependence of $\tau(E) (v(E))^2 g(E)$ Hence the shape of the plots in Figs 1 and 5 depend critically on the choice of energy dependence of $\tau(E) (v(E))^2$. Different choices of $\tau(E) (v(E))^2$ will give very different plots. Appendix A assumes that $\tau(E)$ is independent of E, which seems reasonable. However, the energy-dependence of $v(E)$ is intimately related to $g(E)$, but it also depends on the choice of band-structure (e.g. parabolic bands or something else). Hence it would really help the reader to explicitly explain this, state what assumptions are made (parabolic bands, or something else).

Reply: The information can now be found in the revised Appendix A. Yes, it is true that $Z_{\rm th}T$ being defined from the thermoelastic properties, depends only on the density of states as it characterizes the thermostatic properties of the electron gas, while $z_{\rm e}T$ also depends on $v(E)$ and $\tau(E)$ thus reflecting the transport properties. We wrote a small remark below Eq. (15) to highlight this.

If I assume parabolic bands, then $v(E) \sim E^{1/2}$ (as in H. Ouerdane's response to previous report 2)\ $g(E) \sim E^{(d/2-1)}$ Hence assuming $\tau(E)$ is $E$-independent as stated in appendix A, the term in the integrands for $Z_{e}T$ is $\tau(E) (v(E))^2 g(E) \sim E^{d/2}$, while the equivalent term in integrands for $Z_{th}T$ is $g(E) \sim E^{d/2-1}$. Thus, the integrands in the two quantities always differ by a single power of $E$. Hence, it seems paradoxical that Fig 5 shows a straight line indicating $Z_{e}T=Z_{th}T$ in 1D, when the integrals are clearly different. The authors should explain the details of how that happens.

Reply: We thank the reviewer for this important observation. Indeed, we initially plotted the results for a quantum chain model based on the 0D model. Then we modified the text without updating the figure, hence the confusion. Following this comment, we replaced the previous Fig.~5 with a new one obtained using the 1D electron gas in the Boltzmann approach. The new plot now displays a nonlinear correlation between $Z_{\rm e}T$ versus $Z_{\rm th}T$, as indeed should be expected.

The authors must add the explanation of 0D to the manuscript, the fact it is a quantum dot with a Lorentizian transmission, as explained in the response to the previous report 2.

Reply: The necessary details have been added to the Appendix A.

It is also worth adding a comment to the manuscript that mentions the other difference between $Z_{th}T$ and $Z_{e}T$. The denominator of $Z_{e}T$ contains an extract term proportional the square of the thermoelectric response (the $L_{21}L_{12}$ term in equation for $\kappa_{e}$ in Eq. (31)). There is no analogue of this term in $Z_{th}T$.

Reply: The Onsager kinetic coefficients reflect the transport properties of the electron system in an out-of-equilibrium situation because of, e.g., a temperature difference. Hence they are needed to compute $z_{\rm e}T$. As regards $Z_{\rm th}T$, these coefficients are irrelevant as the thermodynamic figure of merit characterizes the quality of the working fluid, since it is akin to the isentropic expansion factor, which can be defined using the thermoelastic coefficients.

I cannot find the place in the manuscript that gives the value of the electro-chemical potential for the examples in Fig. 4 (these values were given in the response to previous report 2). These values are crucial to understand the curves in Fig 4, because for given $g(E)$ and $v(E)$, the parameter that matters in the ratio of temperature to electro-chemical potential. For example, if one takes a sample with more charge carriers per unit volume so its electro-chemical potential is larger (e.g. larger Fermi energy), then one needs a larger temperature to achieve the same $Z_{e}T$ (and hence achieve the same maximum thermoelectric conversion efficiency).

Reply: For the chemical potentials we used analytical expressions given in Ref. [71]. Note that for the 1D and 0D cases, we used the numerical values obtained for the calculation of the 2D electrochemical potential. Now, as regards the remark on the impact of the carrier concentration on the value of $z_{\rm e}T$, indeed it decreases since with an increase of carrier concentration, the Lorenz number remains nearly constant (because of the increase of electrical conductivity and of the thermal conductivity too) but the Seebeck coefficient (or entropy per particle in thermodynamics) decreases (low carrier concentrations favoring a higher Seebeck coefficient).

Requested changes

Points A and B are strong recommendations, I cannot recommend publication before they are addressed. 2. The other points are optional, but they should be easy to do, and would be a great help to readers.

Reply: We thank the referee for a thorough reading and the useful criticism provided in the report. We have addressed points A and B hoping to clarify any confusing point and showing the added value of our work. We also addressed all the minor points raised in the report, which we treated as seriously as the major ones. We hope we have lifted all possible points of confusion and clarified the rationale of our approach.

The manuscript contains much information and though we endeavored to give a coherent and clear account of the work, we understand that it is not straightforward to read. That said, we hope that the work will attract attention and trigger new experiments as well as (and importantly) theoretical works which will address the problems with less simplifying assumptions, which will allow to go deeper in the physics of thermoelectricity close to a phase transition.

---

## Round 3 · Author Response

Dear Editor, dear Referees,

On behalf of all coauthors, I thank you for your time reviewing our work and the very useful remarks and criticism.

We believe that our manuscript contains much interesting physics about a very challenging problem, and we are grateful for an opportunity to revise our manuscript and resubmit it.

The work is essentially of theoretical nature but we also complemented it with experimental data to make a case for more efforts and attention on the problem of the electron gas fluctuating regimes in thermoelectricity. The manuscript should thus not be viewed as a "theory vs. experiment" report but rather as a work that provides theoretical grounds and experimental data to further explore fluctuating regimes near phase transition and their influence on thermoelectric energy conversion.

Our reply to the Referees' report is provided separately, just below their reports.

Sincerely,
Dr. Henni Ouerdane

---

## Round 3 · List of Changes

The main new change in our work is the discussion on nematic fluctuations to describe the experimental data.
To address the points raised by the Referees we revised our manuscript as follows.

1/ We added text and references on nematic fluctuations in the Introduction;

2/ We clarified the specific aims of our work in the Introduction;

3/ We expanded the Section 2 - Theory, to introduce and explain all the basic ingredients of our thermodynamic model;

4/ We modified our notations for the figures of merit to better distinguish those pertaining to the noninteracting electron gas and those pertaining to the 2D fluctuating Cooper pairs;

5/ We give more detail on the 2D fluctuating Cooper pairs, especially in the new Section 2.2.3;

6/ The new Section 3 is now about our results, numerical and experimental - and it contains parts of the former section 4;

7/ Parts of the former Section 3, which was dedicated to the experimental, notably former sections 3.1 and 3.2, have been moved to the appendix D;

8/ The new Section 4 is dedicated to our discussion, which now includes nematic fluctuations;

9/ The Section Conclusion has been expanded to provide a sharper recap of the work done, and include additional remarks to stress the importance of the fluctuating regimes and briefly indicate the potential for follow-up works as themoelectricity in fluctuating regimes near a phase transition is clearly a path to explore with dedicated experiments and the development of realistic models;

10/ Appendix A has been completed with detail on the parameters used for our numerical simulations.

11/ The bibliography section contains 17 new references.

---

## Editorial Decision

published